# Unsupervised Lifelong Learning with Sustained Representation Fairness

## Abstract

Lifelong learning, pivotal in the incremental improvement of decision-making functions, can be ill-conditioned when dealing with one or several upcoming tasks that insinuate spurious correlations between target labels and sensitive demographic attributes. This often results in biased decisions, disproportionately favoring certain demographic groups. Prior studies to de-bias such learners by fostering fairness-aware, intermediate representations often overlook the inherent diversity of task distributions, thereby faltering in ensuring fairness in a lifelong fashion. This challenge intensifies in the context of *unlabeled* tasks, where discerning distributional shifts for the adaptation of learned fair representations is notably intricate. Motivated by this, we propose *Sustaining Fair Representations in Unsupervised Lifelong Learning* (FaRULi), a new paradigm inspired by human instinctive learning behavior. Like human who tends to prioritize simpler tasks over more challenging ones that significantly outstrip one's current knowledge scope, FaRULi reschedules a buffer of tasks based on the proximity of their fair representations. The learner starts from tasks that share similar fair representations, accumulating essential de-biasing knowledge from them. Once the learner revisits a previously postponed task with more disparate demographic distributions, it is more likely to increment a fair representation from it, as the learner is now provided a larger rehearsal dataset enriched from the learned tasks with diverse demographic patterns. FaRULi showcases promising capability in making fair yet accurate decisions in a sequence of tasks without supervision labels, backed by both theoretical results and empirical evaluation on benchmark datasets. Code is available at: `anonymous.4open.science/r/FaRULi/`.

## 1 Introduction

Organizational decision-making pipelines are increasingly integrating machine learning (ML), with applications ranging from pre-screening eligible engineers among hundreds of thousands of resumes (Dastin, 2018; Bogen & Rieke, 2018), to predicting criminal recidivism (Chouldechova, 2017) or forecasting educational successes (Holstein et al., 2018), to name a few. Yet, the ML algorithms are prone to forming shortcut decision paths (Mehrabi et al., 2021), which spuriously correlate the target variables with the *protected features* of users, such as gender, age, and ethnicity. This could lead to predictive results that are biased against certain demographic groups.

For algorithmic group fairness, previous studies propose to eradicate such spurious correlations through fair representation learning (FRL) (Zemel et al., 2013; Zhang et al., 2018; Chowdhury & Chaturvedi, 2022), de-biasing resultant ML models via learning latent representation that wipes out protected feature information from input. However, the FRL methods mostly focus on mitigating in-domain bias and fail to generalize well to new tasks with data distributional shifts (Barrett et al., 2019). To wit, whereas a FRL-powered resume screening system can make fair decision for similar occupations (e.g., web developer, software engineer, or data scientist), its fairness may wane while screening for very different roles like sales or marketing. Note that in practice hiring tasks are not always predefined and can emerge unexpectedly, e.g., for arising roles during a pandemic break. It is computationally and economically extensive to re-train the FRL model for every new task.

To generalize group fairness across a sequence of diverse tasks, recent advances leverage incremental learning under covariate drift (Zhang & Ntoutsi, 2019; Rezaei et al., 2021; Singh et al., 2021), enabling dynamic model adaptation to evolving data distribution without the necessity of retraining on the entirety of previously seen tasks. Alas, these methods still present two major challenges:

**i**) They suffer from *catastrophic forgetting* (Fatemi et al., 2023), failing to sustain the learned fair representation when adapting to new distributions. For example, then model trained to ensure fair hiring decisions for marketing roles may not maintain its fairness efficacy when applied to engineer hiring scenarios. **ii**) These methods postulate a fully supervised setup and cannot work in label scarce environments, which are a norm in real-world application. In particular, without label in a given task, they cannot gauge the disparity between incoming and previously learned data (Corbett-Davies & Goel, 2018), compromising their capability to discern whether to reuse the existing model or expand it to adapt to new data distributions. Negative model reuse (Chen et al., 2019; He et al., 2021) could occur, if the distributions of the incoming and learned tasks substantially differ.

To overcome the two challenges, we propose a novel learning paradigm, named *Sustaining Fair Representations in Unsupervised Lifelong Learning* (FaRULi). Inspired by human learning instinct (Elman, 1993), our FaRULi approach schedules tasks based on the proximity of their fair representations, allowing the learner to start with tasks that share more similar demographic distribution with the initial task and accumulate de-biasing knowledge of learning fair representations from them. This strategy ensures that when the learner revisits a previously postponed task, it is more likely to sustain the fairness in representation learning because it now has enriched de-biasing knowledge extracted from other prioritized tasks. To realize this intuition, FaRULi leverages an elastic fair representation learning (EFRL) network to map data from the learned and incoming tasks into a shared representation space, in which the task-invariant and label information is preserved while the protected feature information is obfuscated. A key trait of EFRL lies in its adaptive network capacity, which is learned per each incoming task. The larger the distance between a new and the previously learned tasks, the more challenging the extraction of a shared space between them becomes, and consequently, the larger the learning capacity of EFRL is required. As such, EFRL suggests a distance metric capable of assessing task-wise disparity, devoid of assumptions on data distribution or label availability, thereby making FaRULi adaptable over long time spans.

**Specific contributions of this paper are as follows:**

    i) This is the first study to explore fair representation learning from an unsupervised and lifelong perspective, where the learner must induce a shared representation space across a sequence of unlabeled tasks, each characterized by a distinct demographic pattern.

    ii) A novel FaRULi approach is proposed to tackle the problem, anchored in its strategy to reschedule the learning order, prioritizing tasks that exhibit similarities with previously learned tasks, while deferring those that are markedly disparate. FaRULi accumulating knowledge in such order can de-bias the originally diverse demographic distributions without negative model reuse. We detail the technical specifics in Section 2.

    iii) A theoretical analysis on the proposed FaRULi approach is carried out, establishing empirical risk bounds associated with employing a capacity-adaptive learner and optimized task ordering. Findings of this analysis are delineated in Section 3.

    iv) Extensive experiments are conducted to substantiate the viability and effectiveness of our proposal. On average, our model outperforms three state-of-the-art competitors by 12.7% and 42.8% in terms of prediction accuracy and statistical parity, respectively. Detailed results are documented in Section 4.

## 2 THE FARULI APPROACH

**Problem Statement.** We consider a sequence of tasks $\mathcal{T}_i : i = 0, 1, \ldots, N$, where only the first task $\mathcal{T}_0$ has labeled data, and the other tasks $\{\mathcal{T}_i\}_1^N$ remain unlabeled. Define $\mathcal{T}_0 = (X_0, Y_0, P_0)$ and $\mathcal{T}_i = (X_i, P_i), \quad \forall i > 0$, where $X_i \in \mathbb{R}^{|\mathcal{T}_i| \times d}$ represents the $d$-dimensional data instances of the $i$-th task, $Y_0 \in \{0, 1\}^{|\mathcal{T}_0|}$ the true labels of task $\mathcal{T}_0$. Denoted by $P_i \in \{0, 1\}^{|\mathcal{T}_i|}$ the random variable representing a *protected* feature in each task. Without loss of generality, we let the marginals $P(X_i, P_i) \neq P(X_j, P_j)$ for any $i \neq j$, indicating the task disparities. For simplicity, we consider binary $Y$ and $P$, thereby dividing the predicted results into four groups, shown in the figure below.

Our goal is to train a predictive model $f : X \mapsto Y$ across all tasks. At each round, our model learns task $\mathcal{T}_i$ and returns its predicted labels $\widehat{Y}_i$

| | granted $(Y = 1)$ | rejected $(Y = 0)$ |
|---|---|---|
| favored $(P = 1)$ | **FG** | **FR** |
| deprived $(P = 0)$ | **DG** | **DR** |

before moving to the next task. After $N$ rounds, the true labels of all tasks are revealed $\widehat{Y}_1, \ldots, \widehat{Y}_N$, and the model needs to minimize the empirical risk w.r.t. the fairness constraint, defined as:

$$\min \ \mathbb{E}[\widehat{Y}_i \neq f(X_i)], \text{ subject to } |DG/(DG + DR) - FG/(FG + FR)| \leq \epsilon, \qquad (1)$$

where the constraint term coined as *statistical parity* (SP) (Verma & Rubin, 2018) enforces the level of discrimination presented in the model predictions is below threshold $\epsilon$. Note, this constraint can be implemented with various fairness metrics such as *equal opportunity* (Kallus & Zhou, 2018) or *equalized odds* (EO) (Hardt et al., 2016; Alghamdi et al., 2022). Specifically, $EO = \max\{|\,P(\widehat{Y}_i = 1 \mid P = 0, Y) - P(\widehat{Y}_i = 1 \mid P = 1, Y)|, \forall Y\}$ requires the predicted positives ($\widehat{Y}_i = 1$) to be independent of the protected feature $P$ and conditioned on true labels $Y$, which eliminates the negative affect of large number of FR in the dataset. In practice, one can implement the fairness constraint in Eq. (1) differently to better accommodate the application requirements. In this study, both SP and EO are implemented in training and evaluation phases in our experiments.

An **overview** of the objective function empowering our FaRULi approach takes the following form:

$$\mathcal{L}_{\text{EFRL}} = \max_{g,D} \min_{\phi,f} \sum_{l}^{L} \alpha_i^{(l)} (\mathcal{L}_{\text{Pred}}^{(l)}(\phi, f) - \mathcal{L}_{\text{Obfs}}^{(l)}(\phi, g) - \mathcal{L}_{\text{Disc}}^{(l)}(\phi, D)), \tag{2}$$

In Section 2.1, we elucidate how to map data from learned and incoming tasks into a shared latent space to achieve similar data representations by maximizing $\mathcal{L}_{\text{Disc}}$. Fair data representations containing label information are obtained by maximizing $\mathcal{L}_{\text{Obfs}}$ and minimizing $\mathcal{L}_{\text{Pred}}$, respectively. In Section 2.2, we give details on how to build an elastic network such that fair representation learning is conducted in each layer, with weight parameters being updated based on cumulative loss of its corresponding layer. A large distance between the incoming and learned tasks will yield a deep network model, indicating the occurrence of negative model reuse. In Section 2.3, a distance-measurement metric which considers both the depth of model and weight parameters $\alpha$ is introduced to optimize the learning order to avoid negative model reuse. Generating high-confidence pseudo-labels for instances replayed in the following learnings to against catastrophic forgetting is also introduced.

## 2.1 SUSTAINING FAIR REPRESENTATION VIA ADVERSARIAL LEARNING

Traditional fair representation learning approaches (Beutel et al., 2017; Madras et al., 2018) only learned fair data representations which contain label information while eliminating biases arising from the protected feature on one specific task. Our method could learn fair representations from the labeled task, and be reused for incoming tasks by mapping instances from both learned tasks and incoming tasks into a shared latent space. While fair data representations are learned from the labeled task, representations of incoming tasks are approximated to those labeled ones, thus the model presents comparable performance on new tasks. Specifically, given the first labeled task $\mathcal{T}_0$ and any incoming non-labeled task $\mathcal{T}_i$, we start with the initialization of a retained dataset $R^{(i)}$ to include all instances with labels, thus $R^{(0)} := \mathcal{T}_0 = (X_0, Y_0, P_0)$. For an instance $\mathbf{x}$, its task membership is denoted as m, indicating whether $\mathbf{x}$ is from $R^{(0)}$ (i.e., $m = 0$ if $\mathbf{x} \in X_0$), or from $\mathcal{T}_i$ (i.e., $m = 1$ if $\mathbf{x} \in X_i$). For any instance $\mathbf{x}$, if $\mathbf{x} \in R^{(0)}$, the instance is accompanied with a protected feature $p \in P_0$ and true label $y \in Y_0$; otherwise, it conveys the protected feature $p \in P_i$ only. A mapping $\phi$ is learned to project instances from both incoming and learned tasks into a shared $z$-dimensional latent space and generate representations $\mathbf{z} = \phi(\mathbf{x}) \in \mathbb{R}^z$. Given $R^{(0)}$ and $\mathcal{T}_i$, a classifier group is learned, which includes three functions. Namely, 1) $f(\cdot)$ that predicts the label of each instance $\mathbf{x}$, denoted as $\hat{y} = f(\mathbf{z})$, 2) $g(\cdot)$ that infers the demographic group (favored or deprived) that $\mathbf{x}$ belongs to, denoted as $\hat{p} = g(\mathbf{z})$, and 3) $D(\cdot)$ that discriminates $\mathbf{x}$ by its task membership, denoted as $\hat{m} = D(\mathbf{z})$. The fair representation learning loss function is defined as follows:

$$\mathcal{L}_{\text{FRL}} = \max_{h,D} \min_{\phi,f} \left[ \mathcal{L}_{\text{Pred}}(f(\phi(\mathbf{x})), y) - \lambda_1 \mathcal{L}_{\text{Obfs}}(g(\phi(\mathbf{x})), p) - \lambda_2 \mathcal{L}_{\text{Disc}}(D(\phi(\mathbf{x})), m) \right], \tag{3}$$

where $\lambda_1$ and $\lambda_2$ are positive weight parameters to balance the three terms. Defined by $\ell(\cdot, \cdot)$ gauging the loss between the true and predicted variables. Specifically, the first prediction loss term $\mathcal{L}_{\text{Pred}}(\phi, f) = \mathbb{E}_{(\mathbf{x},y) \in R^{(0)}} [\ell(y, \hat{y})]$ is minimized to enhance the capability of $f(\cdot)$ in predicting the label, thus retaining label information in data representations from $R^{(0)}$. The second obfuscation loss term $\mathcal{L}_{\text{Obfs}}(\phi, g) = \mathbb{E}_{(\mathbf{x},p) \in R^{(0)} \cup \mathcal{T}_i} [\ell(p, \hat{p})]$ is maximized to make $g(\cdot)$ fail to infer the demographic group, thus obfuscate protected information from data representations from both tasks. The third task discrimination loss term $\mathcal{L}_{\text{Disc}}(\phi, D) = \mathbb{E}_{(\mathbf{x},m) \in R^{(0)} \cup \mathcal{T}_i} [\ell(m, \hat{m})]$ is also maximized to lower the ability of $D(\cdot)$ to determine whether an instance $\mathbf{x}$ originates from $R^{(0)}$ or $\mathcal{T}_i$. As a result, the learned representations of data are shared by $R^{(0)}$ and $\mathcal{T}_i$, enabling unsupervised model transfer (Bengio, 2012; Long et al., 2016), such that if the learner can make accurate and fair predictions

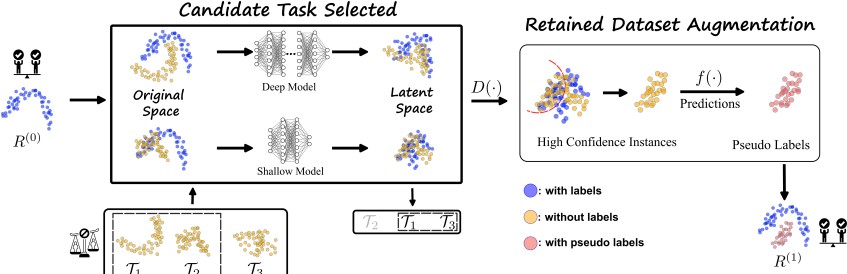

Figure 1: The workflow of our method. $R^{(0)}$ is initialized as the labeled task, and its distances from $\mathcal{T}_1$ and $\mathcal{T}_2$ are gauged by the depths of the learned EFRL networks. $\mathcal{T}_2$ is selected as the candidate because extracting shared representation from $R^{(0)}$ and $\mathcal{T}_2$ requires a shallower network. Instances belonging to $\mathcal{T}_2$ but wrongly classified as part of $R^{(0)}$ by $D(\cdot)$ and their high confidence pseudo-labels generated by $f(\cdot)$ are incorporated into the retained dataset: $R^{(0)} \rightarrow R^{(1)}$, being replayed in the next learning process with $\mathcal{T}_1$ and $\mathcal{T}_3$. Instances with labels are represented by blue points; those without labels are represented by yellow; and those with pseudo-labels are represented by pink.

on the data representations from $R^{(0)}$, comparable performance on those from $\mathcal{T}_i$ can be envisioned.

## 2.2 ELASTIC FAIR REPRESENTATION LEARNING NETWORK

While constructing shared representations for sustaining fairness in two tasks, the fact that any incoming task $\mathcal{T}_i$ has no labeled data make an early-stop of training impossible, increasing the likelihood of *overfitting* on the labeled $R^{(0)}$, thereby incurring negative knowledge transfer (Wang et al., 2019). This issue cannot be observed by evaluating the prediction performance on incoming tasks because of the unavailability of label information. Instead of relying on labels, we propose an EFRL network that enables gauging task-wise distances using its adaptive learning capacity (Ganin & Lempitsky, 2015; Long et al., 2015). The depth of EFRL is deemed as a learnable parameter, and the deeper the model, the farther the distribution of the incoming task is from that of learned tasks. Hence, upon activation of a deep network, we can discern the emergence of negative model reuse.

Specifically, we build an over-complete neural network consisting of $L$ layers and for any $l$-th layer, it is assigned with a weight parameter $\alpha^{(l)}$ and linked with an independent classifier group. Every layer establishes and generates its unique latent space and fair data representations. After every batch training, the losses for each layer are updated via Hedge BackPropagation (HBP) (Freund & Schapire, 1997; Sahoo et al., 2018). The losses and predictions of the EFRL network are determined by a weighted sum of all layers, defined as:

$$\mathcal{L}_{\text{EFRL}} = \min_{\phi,f} \max_{h,D} \sum_{l=1}^{L} \alpha^{(l)} \left[ \mathcal{L}_{\text{Pred}}^{(l)}(\phi, f) - \lambda_1 \mathcal{L}_{\text{Obfs}}^{(l)}(\phi, g) - \lambda_2 \mathcal{L}_{\text{Disc}}^{(l)}(\phi, D) \right],$$

$$\hat{y} = \sum_{l}^{L} \alpha^{(l)} \hat{y}^{(l)}, \hat{p} = \sum_{l}^{L} \alpha^{(l)} \hat{p}^{(l)}, \hat{m} = \sum_{l}^{L} \alpha^{(l)} \hat{m}^{(l)},$$

(4)

where the superscript $l$ indicates that the loss terms are taken over the $l$-th layer. Besides, the loss of each layer is accumulated after $T$ batch trainings to update weight parameters as: $\alpha^{(l)} = \alpha^{(l)} \beta^{\sum_{t=1}^{T} \mathcal{L}_{\text{EFRL}}} / \sum_{l=1}^{L} \alpha^{(l)} \beta^{\sum_{t=1}^{T} \mathcal{L}_{\text{EFRL}}}$, which guarantees $\forall \alpha^{(l)} \in (0, 1)$. Denoted by $\beta$ is the discount rate parameter. The greater the loss suffered by a layer, the smaller its corresponding weight. Hence, the layer with the largest weight value is considered as the largest depth of the model.

The intuitions behind the design of our EFRL network are as follows. First, because of the phenomenon coined *diminishing feature reuse* (Huang et al., 2016; Larsson et al., 2017), where deep representations tend to wash out the semantic meanings of raw feature inputs due to random initialization, the deeper layers converge more slowly than shallower layers. Thus, the weight parameters $\alpha^{(l)}$ associating with shallow layers (i.e., with small $l$) will dominate during the early training phases. Second, as the adversarial training proceeds, deep layers start to take over by gradually increasing the values of their associated weights $\alpha^{(l)}$. This is because that a deep layer $l$ with large learning capacity is likely to yield representation that better obfuscates protected feature information (i.e., enlarging $\mathcal{L}_{\text{Obfs}}^{(l)}(\phi, g)$) and extracts shared inter-task distributions (i.e., enlarging $\mathcal{L}_{\text{Disc}}^{(l)}(\phi, D)$), thereby minimizing the loss $\mathcal{L}_{\text{EFRL}}^{(l)}$ at its corresponding depth $l$, as defined in Eq. (4). Third, the weights

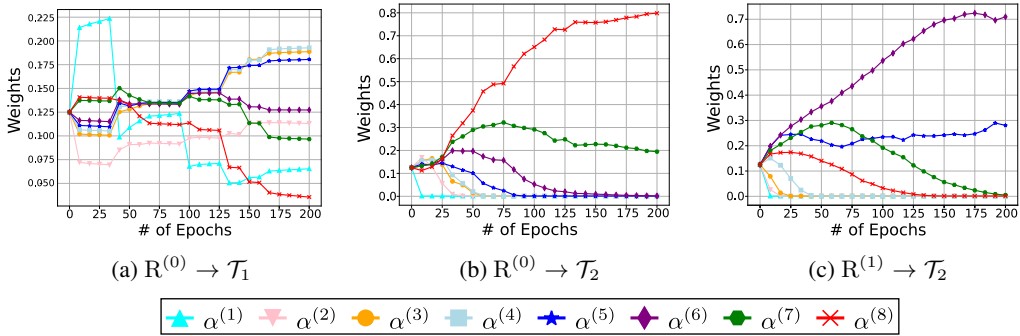

Figure 2: Trends of weight coefficients at various network depths w.r.t. the number of training epochs. Deeper layers tend to dominate if the incoming task is more distant from the retained data.

$\alpha^{(l)}$ of very deep layers (i.e., very large $l$) will remain small after convergence, because such layers have accumulated large loss $\sum_{t=1}^{T} \mathcal{L}_{\text{EFRL}}$ during the previous $T$ training iterations, which lowers the value of $\alpha^{(l)}$ experientially after applying the discount $\beta \in (0, 1)$.

We carry out both theoretical and empirical studies to rationale our design intuition. Specifically, we derive Theorem 1 in Section 3, which suggests the existence of an optimal, intermediate layer $l^\star \in [1, L]$, and that our EFRL network can approximate a network trained with fixed-depth $l^\star$, while knowing the exact value of $l^\star$ across all tasks is impossible in practice. In terms of experiment, we follow the study by He et al. (2021) to visualize the dynamics of layer weights during training, as shown in Figure 2. We reduce the experiments on the Bias-MNIST dataset for visualization, where the updating trends of weights with respect to the number of training epochs are presented. The third task $\mathcal{T}_2$ is deemed as the most faraway task, in which the distribution of background colors over digit types is very different from other four tasks; the details of Bias-MNIST are presented in Section 4.1. We make three observations. First, while the incoming task $\mathcal{T}_1$ is close to the retained dataset $\text{R}^{(0)}$, only shallower layers are enough to approximate the latent space and deeper layers are not activated. Second, negative model reuse is incurred if we enforce learning $\mathcal{T}_2$ from $\text{R}^{(0)}$ immediately. As shown in Figure 2b, even the weights of shallow layers are the same as that of deeper layers at initial iteration, our EFRL network ends up with assigning large weights on deep layers. Third, if we prioritize other tasks by postponing $\mathcal{T}_2$, the shallow layers in our EFRL network would suffice to approximate such a mapping $\phi$ that satisfies Eq. (4), of which the weights remains large as training proceeds, as illustrated in Figure 2c. These observations coincide with our intuition, rationalizing the use of learned weight coefficients in EFRL network to quantify the distance between tasks.

## 2.3 OPTIMIZING LEARNING ORDER AND DATA REHEARSAL

To avoid negative model reuse, the fair representation learning is not conducted as soon as observing a new incoming task. Instead, we select the task which presents the most similar distribution to learned tasks from a set of tasks as the candidate for following learning. The more distant the incoming task from the learned tasks, the deeper the resultant EFRL network. We introduce a similarity-measurement metric which considers both the model depth and weight parameters, defined as $\text{Q} = -\sum_{l=1}^{L} l \cdot \alpha^{(l)} \log \alpha^{(l)}$. The larger value it returns, the further the distribution distance of the incoming task is from learned tasks. As shown in Figure 1, the retained dataset $\text{R}^{(0)}$ is initialized, while two tasks $\mathcal{T}_1$ and $\mathcal{T}_2$ arrives in. The fairness representation learning is conducted for $\mathcal{T}_1$ and $\mathcal{T}_2$ respectively, each alongside $\text{R}^{(0)}$ to obtain a latent space where: 1) data representations of $\text{R}^{(0)}$ are fair; 2) representations of new tasks are similar to those of $\text{R}^{(0)}$. Because $\mathcal{T}_1$ shows a more distinct distribution with $\text{R}^{(0)}$ than $\mathcal{T}_2$, its learning process is at the expense of invoking a deeper model and returns a larger Q value. Hence, $\mathcal{T}_2$ is selected and $\mathcal{T}_1$ remains for the next learning process with a new incoming task $\mathcal{T}_3$.

Identifying the closest task also helps address the catastrophic forgetting in lifelong learning. Because, except for the first task $\mathcal{T}_0$, all incoming tasks are devoid of labels under unsupervised, we use pseudo-labels (predictions made by the predictor $f(\cdot)$) as the replacements of true labels, to replay them alongside with corresponding data during following learning processes. Without optimizing the learning sequence, when the distribution of the incoming task deviates highly from that of learned tasks, the data representation of the new task cannot approximate that of the existing

tasks. This discrepancy will render the predictor $f(\cdot)$ incapable of producing accurate predictions. Therefore, after task re-ordering, we assign pseudo labels to data instances that can be predicted by $f(\cdot)$ with high confidence. We leverage the likelihood of task discrimination as the measurement of prediction confidence. The less likely that the task discriminator $D$ can differentite whether the instance is originally from $\mathrm{R}^{(0)}$ or $\mathcal{T}_2$ after representation, the higher the confidence level of making prediction on this instance. The retained dataset is updated to $\mathrm{R}^{(1)}$ by integrating such instances. A new learning process between $\mathrm{R}^{(1)}$ and two tasks, $\mathcal{T}_1$ and $\mathcal{T}_3$, will be repeated with knowledge from $\mathcal{T}_2$ replayed in the form of pseudo-labeled instances.

## 3 THEORETICAL ANALYSIS

In this section, we investigate the risk bounds of the EFRL network model and the optimized learning order. *First*, in EFRL, we observe that the prediction is based on the outputs of all hidden layers in weighted combination, where several layers may yield less expressive latent representations (*e.g.*, the too shallow or too deep layers) to negatively affect learning performance. Suppose there exists an oracle knowing the optimal depth of the adversarial network for each task beforehand, resulting in global optimum. We firstly investigate the performance gap between EFRL that learns the optimal model depth from data and the oracle-powered model that is always initialized with optimal depth.

**Theorem 1.** *Over $T$ rounds of batch training, our FaRULi suffers cumulative loss:*

$$\mathcal{L}_{\text{FaRULi}} \leq C_\beta \cdot \min_{l^\star} \Big\{ \sum_{t=1}^{T} \mathcal{L}_{\text{EFRL}}^{(l),t} \Big\}_{l=1}^{L} + \frac{\ln L}{1-\beta}, \tag{5}$$

*where $C_\beta = \ln(1/\beta)/(1-\beta) > 0$ is a monotonically decreasing scalar.*

*Remark 1.* A hindsight optimal model of which the *optimal* depth $l^\star$ that yields the least learning loss over $T$ rounds, presented as the above equation, provides a natural upper bound of our FaRULi model. Theorem 1 suggests that our model is comparable to this optimal model (*cf.* $\lim_{\beta \to 1} C_\beta = 1$, $\ln L/(1-\beta) < 0$). As in practice the optimal $l^\star$ is unknown and can vary according to datasets, it is not realistic to conduct a set of experiments to decide the optimal depth for each dataset. Instead, our method can help the model automatically learn the optimal depth at each round and achieve the comparable cumulative loss. Hence, our FaRULi learner strictly enjoys a lower learning loss than any neural network models with their depth fixed in ad-hoc.

*Second*, our FaRULi reschedules the order of learning tasks based on the distance metric suggested by the EFRL model. The theoretical performance of this task order optimization is bounded by:

**Theorem 2.** *Denoted by $\epsilon_{\mathrm{R}^{(i)}}(\hat{h})$ and $\epsilon_{\mathcal{T}_i}(\hat{h})$ the empirical risks suffered by using $\hat{h}$ to predict data in $\mathrm{R}^{(i)}$ and $\mathcal{T}_i$, respectively. Let $\mathcal{H}$ be a hypothesis space on $X$ with VC dimension $d$. $|\mathrm{R}^{(i)}|$ and $|\mathcal{T}_i|$ are samples of size $n$ from two domains $\mathrm{R}^{(i)}$ and $\mathcal{T}_i$ respectively. For any $\delta \in (0, 1)$, with probability at least $1 - \delta$,*

$$\epsilon_{\mathcal{T}_i}(\hat{h}) \leq \epsilon_{\mathrm{R}^{(i)}}(\hat{h}) + \frac{1}{2}\hat{d}_{\mathcal{H}\Delta\mathcal{H}}\left(|\mathrm{R}^{(i)}|, |\mathcal{T}_i|\right) + 4\sqrt{\frac{d\log(2n) + \log\left(\frac{2}{\delta}\right)}{4n}} + \gamma. \tag{6}$$

In Theorem 2, we let $h(\cdot)$ denote the EFRL model learned from the hypothesis space $\mathcal{H}$. The empirical risk for any model can be defined as: $\epsilon_{\mathbb{R}^z}(h, f) = \mathrm{E}_{\mathbf{z} \sim \mathbb{R}^z}[|h(\mathbf{z}) - f(\mathbf{z})|]$, where the error difference between $h(\cdot)$ and any other predictor $f(\cdot)$ is calculated. When $f(\cdot)$ indicates the real data-label distribution, the risk can be abbreviated as $\epsilon_{\mathbb{R}^z}(h)$. In addition, an ideal joint hypothesis $h^*$ from the hypothesis space is the hypothesis which minimizes the combined errors on both retained dataset and candidate task: $h^* = \arg\min_{h \in \mathcal{H}} \epsilon_{\mathrm{R}^{(i)}}(h) + \epsilon_{\mathcal{T}_i}(h)$. The combined error of the ideal hypothesis is defined as: $\gamma = \epsilon_{\mathrm{R}^{(i)}}(h^*) + \epsilon_{\mathcal{T}_i}(h^*)$. Note, no such $h^*$ can be obtained in practice, as $\mathcal{T}_i$ has no label unless $i = 0$. As such, we can remark:

**Remark.** The bound suggested by Theorem 2 establishes a relationship between the empirical risks and the H-divergence of retained dataset and the candidate task. It indicates that the additional risks associated with the elastic model are solely influenced by the distance between the retained dataset and the incoming task. The closer distance between two tasks, the fewer prediction errors the elastic model will incur. This is in accordance with our design that we introduce the task membership discrimination loss $\mathcal{L}_{\text{Disc}}$ which always approximates retained dataset and every incoming task in a shared latent space. Especially for the candidate task, it has the closet distance to the retained dataset, so the least prediction errors will be made while adapting knowledge to it. We deduce that if the distance between distributions of the retained dataset and the candidate task is less than $\frac{1}{2}d_{\mathcal{H}\Delta\mathcal{H}}\left(\mathrm{R}^{(i)}, \mathcal{T}_i\right)$, the empirical risk of the model will be minimized.

## 4 EXPERIMENTS

### 4.1 EVALUATION SETUP

Empirical results are presented to verify the viability and effectiveness of our FaRULi. Detailed experimental setups are elaborated in Section 4.1. Results and findings are extrapolated in Section 4.2.

| No. | Dataset | # Samples | # Features | # Tasks | Rejected / Granted | Deprived / Favored |
|-----|---------|-----------|-----------|---------|--------------------|--------------------|
| 1 | Adult | 30 010 | 15 | 12 | 75:25 | 32:68 |
| 2 | KDD Census-Income | 199 523 | 41 | 9 | 94:6 | 52:48 |
| 3 | Bank marketing | 31 647 | 17 | 12 | 88:12 | 40:60 |
| 4 | Dutch census | 42 125 | 12 | 10 | 52:48 | 50:50 |
| 5 | Diabetes | 71 236 | 50 | 9 | 54:46 | 46:54 |
| 6 | Law School | 14 298 | 23 | 6 | 5:95 | 16:84 |
| 7 | Bias-MNIST | 60 000 | $28 \times 28 \times 3$ | 5 | 10: ...:10 | 68:32 |

### 4.1.1 Dataset Preparation

Our FaRULi methodology is evaluated on seven real-world datasets across four fields, demonstrating its broad applicability. Descriptive statistics related to the datasets under study are provided in above Table. Because propagating knowledge across multiple disparate datasets is a much different research question, we conduct the incremental learning setting within datasets where the features selected for splitting sub-tasks have both direct and indirect correlations with the label and protected feature, respectively, inspired by Le Quy et al. (2022)

● *Financial Datasets (Entries 1-4):* The `Adult` dataset comprises demographic information utilized to predict if an individual has high income. Gender is recognized as a protected feature, and the dataset is segmented into 12 occupation-based tasks. `KDD Census-Income` dataset shares the same prediction task and protected attribute as `Adult`, and is partitioned into 9 tasks according to work-class. `Bank marketing` dataset involves the prediction task forecasting whether a client will subscribe to a deposit scheme. The protected attribute is marital status, and the dataset is divided into 12 occupation-based tasks. The `Dutch census` dataset aims to predict whether an individual is engaged in a high-level occupation. Gender is treated as the protected attribute, and the dataset is partitioned into 10 age-based tasks. ● *Healthcare Dataset (Entry 5):* The `Diabetes` dataset aims to predict the likelihood of a patient being readmitted within 30 days. Gender is treated as the protected attribute, and it is split into 9 age-based tasks. ● *Educational Dataset (Entry 6):* The `Law school` dataset is to anticipate whether a candidate would pass the bar exam. Ethnicity is considered as the protected feature, where white demographic is recognized as favored and other five ethnic groups are classified as deprived. The dataset is divided into six tasks based on Tier. ● *Vision Dataset (Entry 7):* We follow (Bahng et al., 2020) to construct a `Bias-MNIST` dataset consisting of $28 \times 28 \times 3$ hand-writing digits, with background color of each image defining the protected feature. We divide the dataset into five tasks, with each task containing ten digits from 0 to 9. The tasks differ in two aspects: 1) the class distribution $P(Y)$, e.g., $\mathcal{T}_3$ contains the largest number of digit 7 and while $\mathcal{T}_5$ has the least, and 2) the conditional distribution $P(Y|P)$, e.g., $\mathcal{T}_1$ has more than 80% images of digits 1, 2, and 3 in red, while $\mathcal{T}_3$ has more than 80% of them in green.

### 4.1.2 Compared Methods

Five rival models are employed for comparative study. ● *Fair Adversarial Multi Task Learning (FaMTL):* jointly learns all tasks with full label information, rather than employing incremental learning. ● *ULLC (He et al., 2021):* is a lifelong method which only focuses on maximizing accuracy without considering fairness concerns. ● *Fair Adversarial Debiasing Learning (FaDL) (Zhang et al., 2018):* employs a fair adversarial network framework to increase the prediction capability of target labels while reducing the influence of protected feature in the predictions. ● *FaIRL (Chowdhury & Chaturvedi, 2023):* tackles the challenge of incremental learning by incorporating a replay strategy. Parts of samples from previous tasks are randomly sampled for all following trainings to preserve performance across all tasks. It relies on access to complete label information. ● *UnFaIRL:* is a variant of our method. It learns upon the observation of new task instead of re-ordering tasks as FaIRL did, but under unsupervised settings. ULLC and FaDL are designed for one task, and extended to an incremental model with the re-play mechanism for fair comparisons. In this setup, the first task is provided with full label information, while subsequent tasks utilize pseudo-labels as replacements for ULLC, FaDL and FaIRL. The changes in the demographic distribution of each dataset across different tasks can be observed in Figure 3.

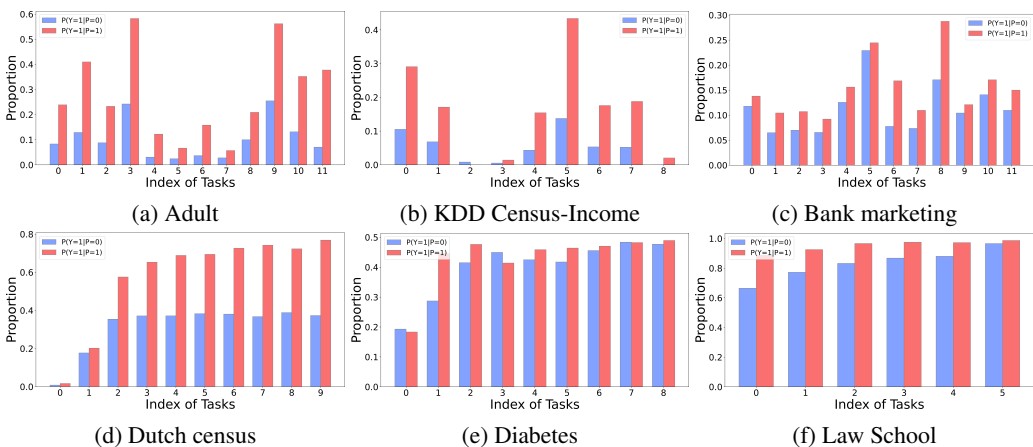

Figure 3: Barchart illustration of the different demographic distributions $P(Y = 1|P = 1)$ and $P(Y = 1|P = 0)$ across all tasks in six studied datasets.

Table 1: Comparative results on 6 datasets with 3 metrics.

| No. | Dataset | FaMTL | ULLC | FaDL | FaIRL | UnFaIRL | FaRULi |
|-----|---------|-------|------|------|-------|---------|--------|
| | | | | Evaluation Metric = Accuracy | | | |
| 1 | Adult | .799 ± .000 | .780 ± .006 | .757 ± .000 | .651 ± .082 | .728 ± .007 | .733 ± .013 |
| 2 | KDD Census-Income | .944 ± .000 | .792 ± .013 | .678 ± .000 | .605 ± .133 | .714 ± .006 | .722 ± .006 |
| 3 | Bank marketing | .891 ± .000 | .839 ± .012 | .684 ± .000 | .580 ± .072 | .722 ± .006 | .730 ± .006 |
| 4 | Dutch census | .789 ± .000 | .784 ± .002 | .526 ± .000 | .472 ± .116 | .747 ± .001 | .752 ± .001 |
| 5 | Diabetes | .618 ± .000 | .565 ± .005 | .459 ± .002 | .501 ± .030 | .584 ± .001 | .590 ± .001 |
| 6 | Law School | .936 ± .000 | .924 ± .006 | .905 ± .000 | .640 ± .088 | .933 ± .001 | .939 ± .002 |
| | | | | Evaluation Metric = Statistical Parity | | | |
| 1 | Adult | .032 ± .000 | .127 ± .038 | .027 ± .000 | .160 ± .095 | .122 ± .023 | .042 ± .007 |
| 2 | KDD Census-Income | .003 ± .000 | .170 ± .016 | .067 ± .000 | .122 ± .093 | .041 ± .011 | .020 ± .003 |
| 3 | Bank marketing | .016 ± .000 | .022 ± .043 | .032 ± .000 | .138 ± .127 | .022 ± .004 | .046 ± .005 |
| 4 | Dutch census | .049 ± .000 | .378 ± .024 | .011 ± .001 | .266 ± .246 | .099 ± .002 | .107 ± .005 |
| 5 | Diabetes | .043 ± .000 | .042 ± .034 | .037 ± .012 | .137 ± .106 | .047 ± .004 | .004 ± .001 |
| 6 | Law School | .016 ± .000 | .149 ± .018 | .071 ± .000 | .225 ± .153 | .034 ± .001 | .002 ± .000 |
| | | | | Evaluation Metric = Equalized Odds | | | |
| 1 | Adult | .163 ± .000 | .214 ± .018 | .150 ± .000 | .196 ± .082 | .210 ± .014 | .178 ± .005 |
| 2 | KDD Census-Income | .220 ± .000 | .256 ± .021 | .087 ± .000 | .168 ± .101 | .163 ± .011 | .077 ± .003 |
| 3 | Bank marketing | .209 ± .000 | .071 ± .040 | .131 ± .000 | .122 ± .094 | .143 ± .005 | .123 ± .008 |
| 4 | Dutch census | .362 ± .000 | .228 ± .011 | .009 ± .000 | .167 ± .155 | .136 ± .001 | .115 ± .007 |
| 5 | Diabetes | .038 ± .000 | .078 ± .029 | .027 ± .001 | .073 ± .032 | .060 ± .001 | .022 ± .002 |
| 6 | Law School | .366 ± .000 | .514 ± .025 | .285 ± .000 | .228 ± .133 | .168 ± .002 | .096 ± .001 |

### 4.1.3 Evaluation Metrics

To evaluate model performance on all tasks, we extend the metric introduced in Section 2 to the accumulative version. The accumulative accuracy refers to the average accuracy (Lopez-Paz & Ranzato, 2017) of the model across all tasks up to and including the current task $\mathcal{T}_N$, defined as Accuracy $= 1/N \sum_i^N \text{Acc}(\mathcal{T}_i)$, where $\text{Acc}(\mathcal{T}_i)$ returns the accuracy on $\mathcal{T}_i$. Similarly, accumulative weighted statistical parity and equalized odds are defined as $\text{SP} = \sum_i^N |\omega_i SP(\mathcal{T}_i)|$, $\text{EO} = \sum_i^N |\omega_i EO(\mathcal{T}_i)|$, $\omega_i = \frac{|\mathcal{T}_i|}{\sum_i^N |\mathcal{T}_i|}$, where $SP(\mathcal{T}_i)$ and $EO(\mathcal{T}_i)$ return the statistical parity and equalized odds on the predicted $\mathcal{T}_i$, respectively, and $|\mathcal{T}_i|$ denotes the number of instances in $\mathcal{T}_i$. For Bias-MNIST that has multiple classes, we follow (Hardt et al., 2016) to take the class that maximizes EO as the calculated EO value. The weight parameters $\omega_i$ represent the proportion of each task's sample size in the entire dataset. Therefore, this weighted sum can prevent the phenomenon where the model performs poorly on tasks with large sample sizes, yet still achieves a fair overall result. The higher the accuracy, the lower the statistical parity, the better the model performs.

### 4.2 RESULTS AND FINDINGS

In an endeavor to mitigate the potential error, each experiment was replicated ten times to present its average value and standard deviation. The collective performance of the algorithm across all datasets is presented in Table 1, whereas performance on every task is delineated in Figure 4, Figure 5 and Figure 6. We present these experimental results to answer two research questions (**Q1 – Q5**).

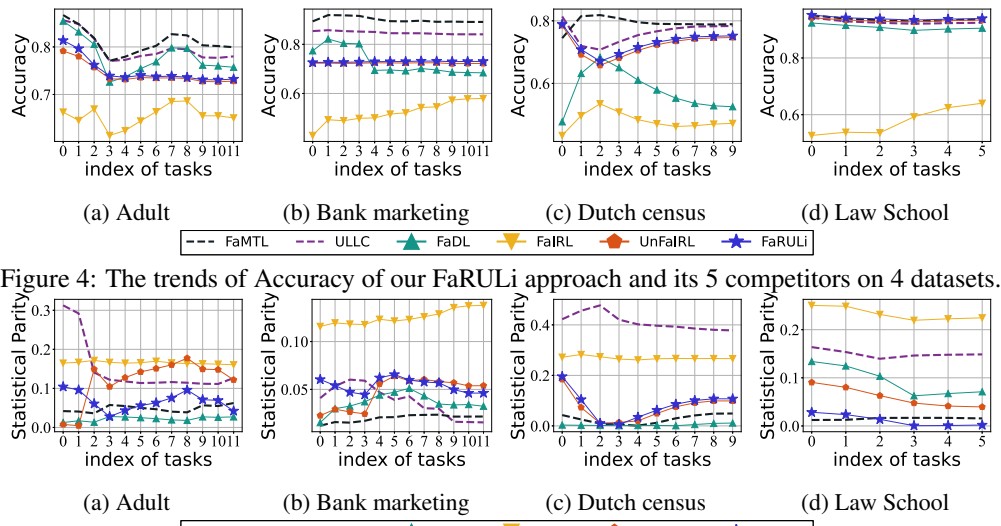

Figure 4: The trends of Accuracy of our FaRULi approach and its 5 competitors on 4 datasets.

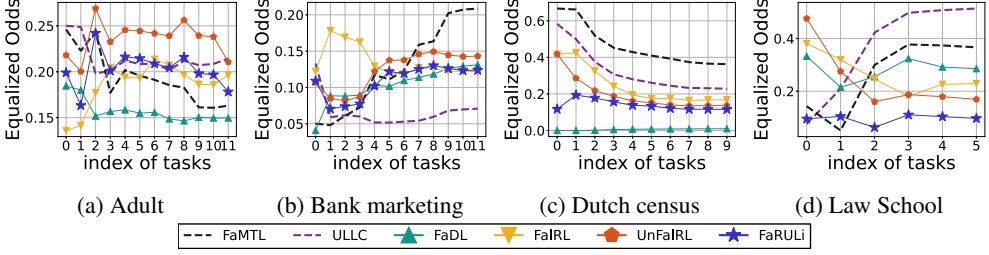

Figure 5: The trends of Statistical Parity of our FaRULi approach and its 5 competitors on 4 datasets.

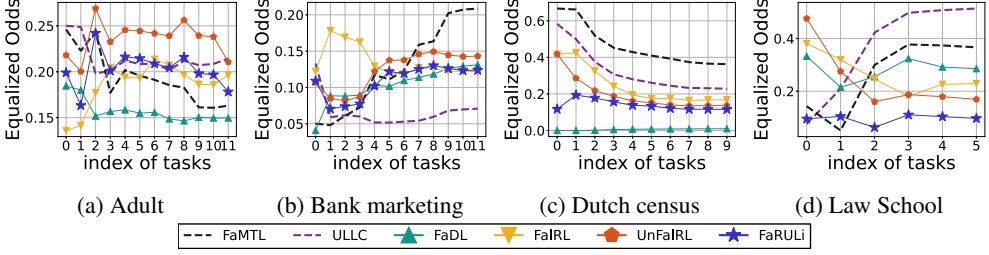

Figure 6: The trends of Equalized Odds of our FaRULi approach and its 5 competitors on 4 datasets.

**Q1. How does our FaRULi compare to the state-of-the-art rival models?**  We will address this question from two perspectives. First, we compare our FaRULi with two upper bound baselines. For FaMTL, it trains all tasks together under full supervision conditions. Hence, it should outperform our algorithm in terms of accuracy and fairness across all datasets. However, as shown in Table 1, Figure 4, and Figure 6 our FaRULi has higher accuracy and lower statistical parity and equalized odds on the `Law School` dataset. In addition, apart from the `KDD Census-income` and `Bank marketing` datasets, the average performance of FaMTL is only 3.2% higher than ours on the remaining four datasets. When comparing statistical parity, we can observe that FaRULi yields smaller values on two of the six datasets: `Diabetes`, and `Law School`. Its average performance is 1.0% higher than FaMTL among all datasets. ULLC has labels only for the first task but does not consider the bias resulting from protected information, so it is also a accuracy upper bound approach. However, it only outperforms FaRULi in accuracy on four datasets. While considering statistical parity and equalized odds, the values of FaRULi are on average 14.5% and 0.125 lower than that of ULLC, respectively. This is particularly evident in the `Dutch` dataset, where the statistical parity of ULLC is 37.8% nearly four times the value obtained by ours. Therefore, our algorithm achieves comparable results in terms of both accuracy and fairness with two upper bound baselines. Second, we compare our FaRULi with two rival models, FaDL and FaIRL. To ensure fair comparative study, pseudo-labeled instances are replayed in both our approach and its competitors in the lifelong learning process. For FaDL, FaRULi demonstrates superior accuracy on 5 out of 6 datasets, exceeding FaDL by over 10% on the `KDD Census-income`, `Dutch`, and `Diabetes` datasets. In terms of statistical parity and equalized odds, FaRULi also obtains better values on 3 and 4 datasets, respectively. FaDL achieves a much smaller statistical parity than ours on the `Dutch` dataset, but it only achieves the accuracy of 52.6% on `Dutch`, and is over 20% lower than ours' 75.2%. We can also observe from Figure 4c, Figure 5c and Figure 6c, FaDL only propagated knowledge related to fairness to the next task but ignored label information. This is because the performance of FaDL is influenced by $\mathcal{T}_1$ and $\mathcal{T}_2$, which are next to $\mathrm{R}^{(0)}$. Without the assistance of shared latent space and

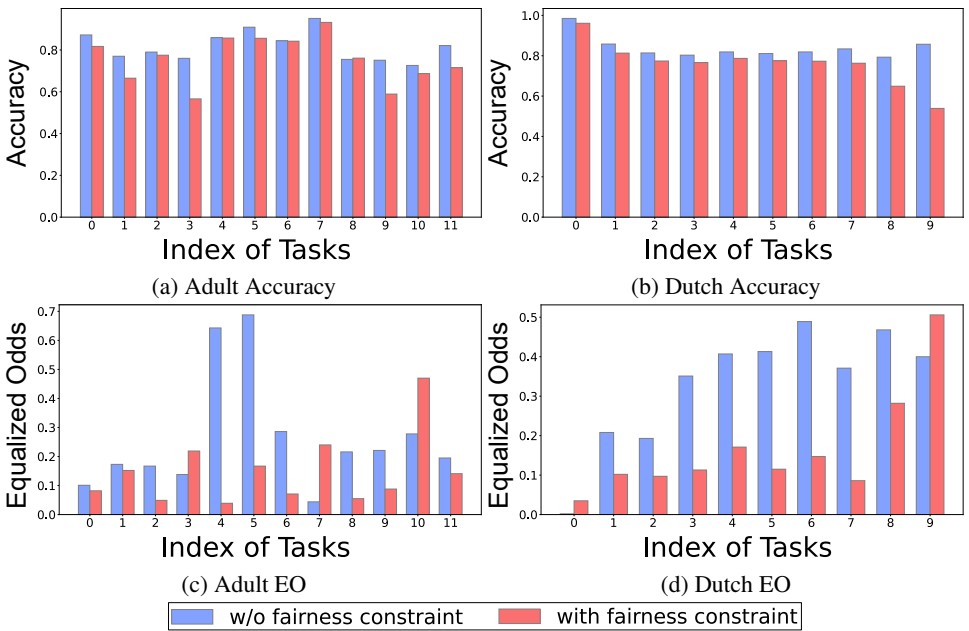

Figure 7: Results of single-task supervised learning with and w/o fairness constraint on `Adult` and `Dutch` datasets.

re-ordering, the forced reuse of the model leads to the bias towards these two tasks, which in turn results in poor performance on the other tasks. Therefore, FaRULi performs better than FaDL in terms of prediction accuracy and fairness. Compared with FaIRL, FaRULi achieves superior performance on 17 out of 18 results. Particular for statistical parity, where values of FaIRL are similar to that of ULLC, which does not exclude protected information. This is due to the instability of this algorithm, a fact confirmed by its high standard deviation.

**Q2. Does lifelong learning with re-ordering lead to performance improvement?** We answer this question by comparing our FaRULi with two rival models, FaDL and FaIRL, and the variant version of our method UnFaIRL. First, although none of the three competitor models possess re-ordering, FaIRL is the worst one, providing high statistical parity and standard deviations on each dataset, observed from Table 1. This instability suggests that, after observing multiple tasks, FaIRL is either unable to stably propagate fair knowledge when encountering new tasks, or it fails to generate accurate pseudo-labels for the new tasks, without re-ordering the learning sequence. Second, FaDL performs better than FaIRL, but still falls short of FaRULi. Our approach outperforms FaDL in five out of the six datasets among both metrics. Particularly for the `Dutch census` dataset, the accuracy of FaDL is only 52.6% while that of FaRULi is 75.2%. We can observe from Figure 4c that the performance of FaDL on each task presents a trend opposite to our method. When observing tasks $\mathcal{T}_1$ and $\mathcal{T}_2$, which show distributions different from other tasks, our method re-orders the learning sequence to learn $\mathcal{T}_1$ and $\mathcal{T}_2$ until the end, while FaDL forces itself to learn these two tasks because they are adjacent to $\mathcal{T}_0$. Although FaDL performs better accuracy on the first four tasks of `Bank marketing`, its result declines after undergoing the learning of $\mathcal{T}_4$, and becomes worse than ours in Figure 4b. Without re-ordering, their learning process cannot avoid the negative reuse cause by $\mathcal{T}_4$. As a result, although our method makes more mistakes during $\mathcal{T}_1$ and $\mathcal{T}_2$, these mistakes do not impact the model's performance on other tasks. In contrast, FaDL's bias towards these two tasks results in poor performance on all other tasks. Final, we employed ablation study to compare FaRULi with its variant UnFaIRL. As evidenced in Table 1, FaRULi outperforms UnFaIRL in terms of two evaluation metrics among most datasets. Specifically, the accuracy achieved by FaRULi is greater than that of UnFaIRL in all datasets, while FaRULi shows smaller statistical parity than UnFaIRL in 4 out of 6 datasets. This difference was especially pronounced within the `Diabetes` and `Law School` datasets, with FaRULi presenting values of 0.4% and 0.2% respectively, as opposed to the 4.7% and 3.4% demonstrated by UnFaIRL. Hence, we can conclude that FaRULi achieves better performance in terms of both accuracy and fairness with the help of

Table 2: Comparative results on Bias-MNIST with 2 metrics.

| Model | $\mathcal{T}_0$ | $\mathcal{T}_1$ | $\mathcal{T}_2$ | $\mathcal{T}_3$ | $\mathcal{T}_4$ | Avg. |
|-------|------|------|------|------|------|------|
| **Evaluation Metric = Accuracy** | | | | | | |
| FaMTL | $.969 \pm .100$ | $.966 \pm .090$ | $.971 \pm .092$ | $.975 \pm .136$ | $.983 \pm .094$ | $.973 \pm .087$ |
| ULLC | $.929 \pm .057$ | $.904 \pm .082$ | $.887 \pm .124$ | $.925 \pm .070$ | $.920 \pm .073$ | $.913 \pm .062$ |
| FaDL | $.939 \pm .089$ | $.911 \pm .078$ | $.855 \pm .085$ | $.851 \pm .089$ | $.886 \pm .065$ | $.888 \pm .066$ |
| UnFaIRL | $.933 \pm .062$ | $.888 \pm .107$ | $.889 \pm .072$ | $.896 \pm .087$ | $.869 \pm .066$ | $.895 \pm .078$ |
| FaRULi | $.952 \pm .125$ | $.929 \pm .096$ | $.887 \pm .100$ | $.946 \pm .110$ | $.933 \pm .079$ | $.929 \pm .088$ |
| **Evaluation Metric = Equalized Odds** | | | | | | |
| FaMTL | $.041 \pm .087$ | $.094 \pm .104$ | $.488 \pm .076$ | $.043 \pm .087$ | $.029 \pm .083$ | $.139 \pm .043$ |
| ULLC | $.212 \pm .081$ | $.361 \pm .079$ | $.489 \pm .107$ | $.247 \pm .069$ | $.243 \pm .051$ | $.310 \pm .110$ |
| FaDL | $.267 \pm .073$ | $.217 \pm .121$ | $.396 \pm .065$ | $.379 \pm .064$ | $.334 \pm .120$ | $.319 \pm .092$ |
| UnFaIRL | $.177 \pm .070$ | $.381 \pm .109$ | $.496 \pm .130$ | $.237 \pm .094$ | $.374 \pm .095$ | $.333 \pm .098$ |
| FaRULi | $.093 \pm .116$ | $.164 \pm .091$ | $.218 \pm .144$ | $.099 \pm .076$ | $.130 \pm .066$ | $.141 \pm .100$ |

re-ordering the learning sequence than three rival models in general. The analysis about under what circumstances re-ordering will lead to substantial improvements is illustrated in the supplementary.

**Q3. What is the upper performance bounds for single-task supervised unconstrained and fairness-constrained baselines?** To better observe the upper bound of the model's ability in prediction and ensuring fairness in our problem, we conducted a set of experiments under full supervision. Specifically, the model is required to perform single-task learning on each task of every dataset, under two conditions: with and without fairness constraints. The results of the experiments on the `Adult` and `Dutch` datasets are presented in the form of barchart in Figure 7. In general, across all tasks in the `Adult` dataset, except for $\mathcal{T}_8$, the model unconstrained by fairness outperforms in terms of accuracy, showing an average improvement of 6.2%. On the `Dutch` dataset, this model achieves superior accuracy on every task, with an average enhancement of 7.9%. Conversely, the model constrained by fairness shows superior performance in fairness metrics EO. It achieves an average improvement of 0.115 and 0.116 on the `Adult` and `Dutch` datasets, respectively. Especially for $\mathcal{T}_4$ and $\mathcal{T}_5$ of the `Adult` dataset, it improves by 0.604 and 0.521, respectively.Besides, for single-task supervised learning without fairness constraints, we can observe that although tasks all originate from the same dataset, their respective prediction accuracies vary largely. For example, in the `Adult` dataset, the highest accuracy is found in $\mathcal{T}_7$, as high as 95.1%, but in $\mathcal{T}_{10}$, the accuracy drops by more than twenty percent to 72.6%. These results substantiate the variation of conditional probability $P(Y|X)$ across different tasks.For single-task supervised learning with fairness constraints, we can observe the large variations in terms of both accuracy and fairness performances. For example, in the `Adult` dataset, we can observe predictions being both fair and accurate (e.g., $\mathcal{T}_0$, acc=81.7% and EO=0.082), fair yet inaccurate (e.g., $\mathcal{T}_9$, acc=58.9%, EO=0.088), accurate yet unfair ($\mathcal{T}_7$, acc=93.2%, EO=0.24), and unfair and inaccurate (e.g., $\mathcal{T}_3$, acc=56.6%, EO=0.22). These results demonstrate the variations of both conditionals in$P(Y|X)$ and $P(Y|P)$, highlighting the performance bound in each single task and calling for lifelong learners that can respect the different variation patterns of $P(Y|X)$ and $P(Y|P)$ across tasks.

**Q4. Can our proposed method be generalized to non-linear high-dimensional datasets?** We leverage the results from Bias-MNIST to answer this question, documented in Table 2 reduced from Table 1 because FaIRL is tailored for binary problem. We present the performance of five models on each task and their average values. The average result is equivalent to the accumulative result, as each task in the Bias-MNIST dataset is of the same scale. From Table 2, we can have two observations. First, our proposed FaRULi is effective on non-linear high-dimensional image datasets as well, as our model achieved comparable performance to FaMTL with fully supervision, evidenced by average accuracy of 92.9% and EO value of 0.141. Compared to the other three algorithms, our method showed an average improvement of 3.3% percent in accuracy and 0.179 in EO.
Second, we observe that all models perform the worst on $\mathcal{T}_2$. Notably, FaMTL achieves EO values below 0.1 in all tasks except for $\mathcal{T}_2$, where the value is 0.488. This is because, here, we intentionally appoint $\mathcal{T}_2$ as the most faraway task, in which its conditional $P(Y|P)$ largely differs from all other four tasks. Therefore, both FaDL and UnFaIRL that learn tasks based on the order of observation

Table 3: Impact of the fairness regularization constants $\lambda_1$ on two datasets.

| Dataset | Metric | $\lambda_1$=0.01 | $\lambda_1$=0.03 | $\lambda_1$=0.05 | $\lambda_1$=0.1 | $\lambda_1$=0.5 | $\lambda_1$=1 |
|---|---|---|---|---|---|---|---|
| Law School | Acc | $.916 \pm .001$ | $.939 \pm .002$ | $.899 \pm .002$ | $.941 \pm .001$ | $.047 \pm .000$ | $.047 \pm .000$ |
| | SP | $.162 \pm .003$ | $.002 \pm .003$ | $.054 \pm .001$ | $.012 \pm .000$ | $.000 \pm .000$ | $.000 \pm .000$ |
| | EO | $.269 \pm .004$ | $.087 \pm .001$ | $.107 \pm .002$ | $.050 \pm .001$ | $.000 \pm .000$ | $.000 \pm .000$ |
| – | – | $\lambda_1$=0.01 | $\lambda_1$=0.03 | $\lambda_1$=0.05 | $\lambda_1$=0.07 | $\lambda_1$=0.09 | $\lambda_1$=1 |
| Bank marketing | Acc | $.742 \pm .004$ | $.740 \pm .009$ | $.730 \pm .006$ | $.727 \pm .008$ | $.714 \pm .005$ | $.661 \pm .012$ |
| | SP | $.071 \pm .004$ | $.010 \pm .002$ | $.046 \pm .005$ | $.118 \pm .004$ | $.167 \pm .004$ | $.575 \pm .026$ |
| | EO | $.053 \pm .006$ | $.038 \pm .005$ | $.064 \pm .002$ | $.142 \pm .008$ | $.189 \pm .005$ | $.650 \pm .053$ |

experience a sharp decline in performance after $\mathcal{T}_2$. For example, FaDL works better in the first two tasks $\mathcal{T}_0$ and $\mathcal{T}_1$ (average accuracy = 92.5%, average EO = 0.242), but cannot generalize to other tasks. In contrast, our model maintains good performance on $\mathcal{T}_0$, $\mathcal{T}_1$, $\mathcal{T}_3$, and $\mathcal{T}_4$, all of which have different conditionals $P(Y|P)$, outperforming FaDL and UnFaIRL by 3.8% and 0.185 in Accuracy and EO on average, respectively. Only on the most faraway $\mathcal{T}_2$, our model arrives at the comparable accuracy performance as them. This aligns with our assumption that the model cannot generalize to a highly disparate task, and our model resolves the negative transfer issue by prioritizing $\mathcal{T}_1$, $\mathcal{T}_3$, and $\mathcal{T}_4$ and achieving significantly better EO performance on them over FaDL and UnFaIRL.

**Q5. What is the impact of the fairness regularization constants $\lambda_1$?** We have conducted the experiments and presented results from two datasets `Bank marketing` and `Law School` in Table 3 for demonstration. For the `Law School` dataset, we observed that as the fairness regularization coefficient $\lambda_1$ increased, SP and EO values appropriately decreased from 0.071 and 0.053 to 0. However, this was accompanied by a decline in accuracy, f rom 91.6% to 4.7%. This pattern highlights the trade-off between accuracy and fairness in lifelong learning settings, suggesting that imposing stringent fairness constraints on the learner might lead to a significant drop in classification accuracy. In such cases, the learner tends to rejects all instances into Y=0, effectively ceasing to make predictions to avoid unfair outcomes.
Another negative trend is evident in the `Bank marketing` dataset. Initially, similar to `Law School`, an increase in $\lambda_1$ from 0.01 to 0.03 causes both SP and EO to decrease, at the expense of reduced accuracy, from 0.071 and 0.053 to 0.010 and 0.038, respectively. However, as the fairness requirement is further heightened, the learner's performance deteriorates in both accuracy and fairness metrics. A distinct increase in SP and EO is observed as $\lambda_1$ rises from 0.05 to 0.09, with values escalating more than threefold, from 0.046 to 0.167 and 0.064 to 0.189, respectively. When $\lambda_1$ finally reaches 1, unlike in the `Law School` dataset, the fairness metrics do not drop to zero. Instead, they continue to increase, surpassing 0.5. These findings suggest that increasing the fairness demands placed on the model can lead to its failure. The model not only shows poor performance by rejecting all points but also becomes ineffective in terms of both prediction accuracy and fairness.

## 5 RELATED WORK

**Fair Representation Learning (FRL).** Whereas bias in learning systems may exsit in various forms and stages, including data, algorithm, and user interaction, FRL methods mainly focus on building fair predictive models from datasets with inherent bias Mehrabi et al. (2021). Preprocessing techniques, e.g., removing protected features, cannot directly eliminate bias from training data (Pedreshi et al., 2008), as the protected feature information could be recovered from other non-sensitive proxy features (Barocas & Selbst, 2016). FRL resolves this issue by learning a latent representation from input data that reduces its statistical parity, firstly introduced by Zemel et al. (2013). Later study explores variational autoencoder (VAE) with a Maximum Mean Discrepancy penalty by deeming protected information as nuisance features (Louizos et al., 2015), which motivates a flurry of VAE variants for FRL (Moyer et al., 2018; Jaiswal et al., 2018; Amini et al., 2019). Paralleling the research effort, Beutel et al. (2017); Madras et al. (2018); Xu et al. (2018); Li et al. (2018); Zhang et al. (2018); Elazar & Goldberg (2018) and Basu Roy Chowdhury et al. (2021) leverage generative adversarial network (GAN) to obfuscate the protected features while maintaining the prediction accuracy on labels. Alongside the group fairness, another research thrust strives to ensure that similar individuals are treated similarly, even if their protected attributes differ, striving to realize counterfactual fairness (Kusner et al., 2017; Dixon et al., 2018; Cheng et al., 2021). While

the FRL frameworks for group fairness demonstrate promising results in a specific domain, they fall short in maintaining fairness when applied to out-of-distribution data (Barrett et al., 2019). As a result, the prior FRL studies mostly cannot work well in a lifelong manner, where a sequence of tasks with highly disparate data distributions are presented.

**Lifelong and Continual Learning.** Lifelong learning aims to build ML systems that can learn a task sequence incrementally (Kirkpatrick et al., 2017; Li & Hoiem, 2017; Rolnick et al., 2019; Hao et al., 2013; Mitchell et al., 2018; Abujabal et al., 2018). The challenge is to overcome catastrophic forgetting, which occurs when new knowledge disrupts previous learned information. In general, existing methods fall into two categories that reuse the model and expand the model. First, there are two main ideas on reusing the model: 1) regularization-based methods, where the model parameters are regularized to avoid drastic updates by limiting the learning rate on important parameters for previous tasks, striving to search a Pareto-effective solution that performs satisfactorily for all seen tasks, relaxing catastrophic forgetting (Kirkpatrick et al., 2017; Aljundi et al., 2017; Shmelkov et al., 2017; Li & Hoiem, 2017; Aljundi et al., 2018). 2) rehearsal-based methods, where parts of instances from previous tasks are stored in an external memory (*i.e.*,the retained dataset) and will be jointly trained along with instances in the current task (Gepperth & Karaoguz, 2016; Schaul et al., 2016; Rebuffi et al., 2017; Lopez-Paz & Ranzato, 2017; Rolnick et al., 2019; Hayes et al., 2019). Second, architecture-based methods are designed to expand the model by either expanding the size of networks (Li et al., 2019; Rao et al., 2019; Zhao et al., 2022), or designing sub-networks for each specific task (Ke et al., 2020; Mallya & Lazebnik, 2018; Serra et al., 2018; Wang et al., 2020). Unfortunately, current methods for lifelong learning focus on the performance of label predictions without considering the model fairness. We note a recent study by Chowdhury & Chaturvedi (2023) that strived to deal with both issues using data rehearsal. However, it postulates a supervised setup where all tasks come with fully labeled instances. Our FaRULi lifts this assumption by requiring label from one initial task only, making the learning scenario less costly and more widely applicable.

## 6 CONCLUSION

This paper presents FaRULi, a novel approach for unsupervised lifelong learning that consistently maintains fair representation across a task sequence. FaRULi overcomes the dual challenges of incrementally adapting fair representations to new tasks and circumventing negative model reuse during distribution shifts in scenarios lacking labels. The crux of our approach lies in a strategic task re-ordering approach, powered by an over-complete elastic model. This model adeptly prevents negative reuse to divergent distributions by implicitly computing task distribution distance without full supervision. We employ an adversarial network framework to create a robust retained dataset of fair data representations, effectively eradicating biases from protected features while preserving label information. The learning task sequence is optimized to re-order based on the calculated distance, selecting the task with minimal deviation from the current retained dataset as the candidate. These tasks are mapped into a shared latent space to generate analogous data representations, with samples from the candidate task supplemented to the retained dataset using high-confidence pseudo-labels. This process ensures efficient subsequent adaptations in a replay manner and sustains the model's overall performance across all tasks. Both theoretical analysis and empirical results from seven real-world datasets substantiate the effectiveness and viability of FaRULi, affirming its potential in fostering fair representations in unsupervised lifelong learning.

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
