# Unsupervised Lifelong Learning with Sustained Representation Fairness

This file is used to supplement the main paper and has three sections. First, we give detailed proof of two theorems proposed in the main paper in Section 1. Second, all experimental results not presented in the main body because of limited pages are shown in Section 2. Third, we have introduced the importance of the re-ordering function in our method. In Section 3, we will further discuss that under what circumstances the re-ordering will lead more substantial improvements. We will analysis this question with the `Law School` dataset. In Section 4, we give implementation details.

## 1 Proof of Theorem

### 1.1 Proof of Theorem 1

**Theorem 1.** *Over $T$ rounds of batch training, our FaRULi suffers cumulative loss:*

$$\mathcal{L}_{\text{FaRULi}} \leq C_\beta \cdot \min_{l^\star} \Big\{ \sum_{t=1}^{T} \mathcal{L}_{\text{EFRL}}^{(l),t} \Big\}_{l=1}^{L} + \frac{\ln L}{1 - \beta}, \tag{1}$$

*where $C_\beta = \ln(1/\beta)/(1 - \beta) > 0$ is a monotonically decreasing scalar.*

***Proof:*** Before our analysis, two loss formulas need to be introduced. For the $l$-th layer, its cumulative loss over T rounds is defined as: $\mathcal{L}_{\text{EFRL}}^{(l)} = \sum_{t=1}^{T} \ell_t^{(l)}$, and the loss is denoted as $\mathcal{L}^{(l)}$ in the follows for simplicity. Similarly, let $\mathcal{L}_{\text{FaRULi}} = \sum_{t=1}^{T} \sum_{l=1}^{L} p_t^{(l)} \ell_t^{(l)} = \sum_{t=1}^{T} \mathbf{p}_t \cdot \ell_t$ as the total cumulative loss suffered by our algorithm. Different from the formal content, a new parameter $p$ is introduced to represent a normalization function, where $\mathbf{p}_t$ is a vector including parameters $p$ of all layers, and allocated by corresponding weight parameter $\alpha_t = [\alpha_t^{(1)}, \ldots, \alpha_t^{(L)}]^\top$. In particular, the relationship between them and the updating rule of parameter $\alpha$ is described as:

$$\alpha_{t+1}^{(l)} = \alpha_t^{(l)} \cdot \beta^{\ell_t^{(l)}}, \quad \mathbf{p}_t = \frac{\alpha_t}{\sum_{l=1}^{L} \alpha_t^{(l)}}. \tag{2}$$

To conduct further proof, we here introduce:

**Lemma 1.** *(Freund & Schapire, 1997) $\beta^r \leqslant 1 - (1 - \beta)$, for $\beta \geqslant 0$ and $r \in [0, 1]$.*

Then, combined with Eq. (2) and Lemma 1, this implies

$$\sum_{l=1}^{L} \alpha_{t+1}^{(l)} = \sum_{l=1}^{L} \alpha_t^{(l)} \beta_i^{\ell_t^{(l)}},$$

$$\leqslant \sum_{l=1}^{L} \alpha_t^{(l)} \left( 1 - (1 - \beta)\ell_t^{(l)} \right), \tag{3}$$

$$= \Big( \sum_{l=1}^{L} \alpha_t^{(l)} \Big) (1 - (1 - \beta)\mathbf{p}_t \cdot \ell_t).$$

Applying repeatedly for $t = 1, \ldots, T$ yields

$$\sum_{l=1}^{L} \alpha_{T+1}^{(l)} \leqslant \prod_{t=1}^{T} \left(1 - (1-\beta)\mathbf{p}_t \cdot \ell_t\right),$$

$$\leqslant \exp\left(-(1-\beta)\sum_{t=1}^{T} \mathbf{p}_t \cdot \ell_t\right),$$

$$= \exp\left(-(1-\beta)\mathcal{L}_{\text{FaRULi}}\right),$$

since $1 + x \leqslant e^x$ for all $x$ and $\mathcal{L}_{\text{FaRULi}} = \sum_{t=1}^{T} \mathbf{p}_t \cdot \ell_t$.

Then, we get the follow formula,

$$\ln\left(\sum_{l=1}^{L} \alpha_{T+1}^{(l)}\right) \leqslant \ln\exp\left(-(1-\beta)\mathcal{L}_{\text{FaRULi}}\right),$$

$$\mathcal{L}_{\text{FaRULi}} \leqslant \frac{-\ln\left(\sum_{l=1}^{L} \alpha_{T+1}^{(l)}\right)}{1-\beta}. \tag{4}$$

Next, going back to Eq. (2),

$$\alpha_{T+1}^{(l)} = \alpha_1^{(l)} \prod_{t=1}^{T} \beta^{\ell_t^{(l)}},$$

$$= \alpha_1^{(l)} \beta^{\mathcal{L}^{(l)}}, \tag{5}$$

and for all layers, we get,

$$\sum_{l=1}^{L} \alpha_{T+1}^{(l)} = \sum_{l=1}^{L} \alpha_1^{(l)} \beta^{\mathcal{L}^{(l)}},$$

$$\geqslant \beta^{\max_l \in L \mathcal{L}^{(l)}} \sum_{l=1}^{L} \alpha_1^{(l)}. \tag{6}$$

By now, all preparations for analyzing Theorem 1 are complete. Combined Eq. (5) and Eq. (6),

$$\mathcal{L}_{\text{FaRULi}} \leqslant \frac{-\ln\left(\sum_{l \in L} \alpha_1^{(l)}\right) - (\ln\beta)\max_{l \in L} \mathcal{L}^{(l)}}{1-\beta}. \tag{7}$$

This is a general bound statement where all layers are be considered. For any $l \in \{1, \dots, L\}$, we achieve a special case:

$$\mathcal{L}_{\text{FaRULi}} \leqslant \frac{-\ln\alpha_1^{(l)} - \mathcal{L}^{(l)}\ln\beta}{1-\beta}. \tag{8}$$

The bound 8 state that our FaRULi only perform a little bit worse than the best $l$-th layer among the sequence. The difference lies in the choice of $\beta$ and the initial weight $\alpha_1^{(l)}$ of each layer. If every weight is set equally such that $\alpha_1^{(l)} = 1/L$, then this bound becomes:

$$\mathcal{L}_{\text{FaRULi}} \leqslant \frac{\min_l \mathcal{L}^{(l)}\ln(1/\beta) + \ln L}{1-\beta}. \tag{9}$$

The bound given in Eq. (9) can be written as:

$$\mathcal{L}_{\text{FaRULi}} \leq C_\beta \cdot \min_{l^\star}\left\{\sum_{t=1}^{T} \mathcal{L}_{\text{EFRL}}^{(l),t}\right\}_{l=1}^{L} + \frac{\ln L}{1-\beta},$$

where $C_\beta = \ln(1/\beta)/(1-\beta) > 0$ as stated in Theorem 1. $\qquad\square$

## 1.2 PROOF OF THEOREM 2

**Theorem 2.** *Denoted by $\epsilon_{R^{(i)}}(\hat{h})$ and $\epsilon_{\mathcal{T}_i}(\hat{h})$ the empirical risks suffered by using $\hat{h}$ to predict data in $R^{(i)}$ and $\mathcal{T}_i$, respectively. Let $\mathcal{H}$ be a hypothesis space on $X$ with VC dimension $d$. $|R^{(i)}|$ and $|\mathcal{T}_i|$ are samples of size $n$ from two domains $R^{(i)}$ and $\mathcal{T}_i$ respectively. For any $\delta \in (0,1)$, with probability at least $1 - \delta$,*

$$\epsilon_{\mathcal{T}_i}(\hat{h}) \leq \epsilon_{R^{(i)}}(\hat{h}) + \frac{1}{2}\hat{d}_{\mathcal{H}\Delta\mathcal{H}}\left(|R^{(i)}|, |\mathcal{T}_i|\right) + 4\sqrt{\frac{d\log(2n) + \log\left(\frac{2}{\delta}\right)}{4n}} + \gamma. \tag{10}$$

***Proof:*** To proof our theorem, we first introduce the triangle inequality for classification error (Ben-David et al., 2006; Crammer et al., 2008) which implies that $\epsilon(h_1, h_2) \leq \epsilon(h_1, h_3) + \epsilon(h_2, h_3)$. Then, we have:

$$\begin{aligned}
\epsilon_{\mathcal{T}_i}(h) &\leq \epsilon_{\mathcal{T}_i}(h^*) + \epsilon_{\mathcal{T}_i}(h, h^*), \\
&= \epsilon_{\mathcal{T}_i}(h^*) + \epsilon_{\mathcal{T}_i}(h, h^*) \\
&\quad + \epsilon_{R^{(i)}}(h, h^*) - \epsilon_{R^{(i)}}(h, h^*), \\
&\leq \epsilon_{\mathcal{T}_i}(h^*) + \epsilon_{R^{(i)}}(h, h^*) \\
&\quad + |\epsilon_{\mathcal{T}_i}(h, h^*) - \epsilon_{R^{(i)}}(h, h^*)|.
\end{aligned} \tag{11}$$

To proceed with the proof, we adapt the definition and inequality suggested by Ben-David et al. (2010) as follows:

**Definition 1.** *For a hypothesis space $\mathcal{H}$, the* symmetric difference hypothesis space $\mathcal{H}\Delta\mathcal{H}$ *is the set of hyperspheres*

$$g \in \mathcal{H}\Delta\mathcal{H} \iff g(\mathbf{x}) = h(\mathbf{x}) \oplus h'(\mathbf{x}) \quad \text{for some } h, h' \in \mathcal{H},$$

*where $\oplus$ is the XOR function, determining whether the outcomes of two functions $h$ and $h'$ are equal.*

If the maximum discrepancy between two functions across two domains are founded, then this value defines the H-divergence distance of two domains as follows:

**Lemma 2.** *For any hyperspheres $h, h' \in \mathcal{H}$,*

$$|\epsilon_{R^{(i)}}(h, h') - \epsilon_{\mathcal{T}_i}(h, h')| \leq \frac{1}{2}d_{\mathcal{H}\Delta\mathcal{H}}\left(\mathcal{T}_i, R^{(i)}\right).$$

So, by Lemma 2, we have:

$$\begin{aligned}
\epsilon_{\mathcal{T}_i}(h) &\leq \epsilon_{\mathcal{T}_i}(h^*) + \epsilon_{R^{(i)}}(h, h^*) + \frac{1}{2}d_{\mathcal{H}\Delta\mathcal{H}}(R^{(i)}, \mathcal{T}_i), \\
&\leq \epsilon_{\mathcal{T}_i}(h^*) + \epsilon_{R^{(i)}}(h) + \epsilon_{R^{(i)}}(h^*) \\
&\quad + \frac{1}{2}d_{\mathcal{H}\Delta\mathcal{H}}(R^{(i)}, \mathcal{T}_i), \\
&= \epsilon_{R^{(i)}}(h) + \frac{1}{2}d_{\mathcal{H}\Delta\mathcal{H}}(R^{(i)}, \mathcal{T}_i) + \gamma.
\end{aligned} \tag{12}$$

With adapting Lemma 3 proposed by Ben-David et al. (2010), the H-divergence distance between two domains $\mathcal{T}_i$ and $R^{(i)}$ can be estimated using a finite number of samples extracted from each domain as follows:

**Lemma 3.** *Let $\mathcal{H}$ be a hypothesis space on $\mathcal{X}$ with $VC$ dimension $d$. If $|R^{(i)}|$ and $|\mathcal{T}_i|$ are samples of size $n$ from two domains $R^{(i)}$ and $\mathcal{T}_i$ respectively and $d_{\mathcal{H}}\left(|R^{(i)}|, |\mathcal{T}_i|\right)$ is the empirical $\mathcal{H}$-divergence between samples, then for any $\delta \in (0,1)$, with probability at least $1 - \delta$,*

$$d_{\mathcal{H}}\left(R^{(i)}, \mathcal{T}_i\right) \leq d_{\mathcal{H}}\left(|R^{(i)}|, |\mathcal{T}_i|\right) + 4\sqrt{\frac{d\log(2n) + \log\left(\frac{2}{\delta}\right)}{n}}.$$

combining Lemma 3 with Eq. (12), we arrive at:

$$\epsilon_{\mathcal{T}_i}(h) \leq \epsilon_{R^{(i)}}(h) + \frac{1}{2}d_{\mathcal{H}\Delta\mathcal{H}}\left(|R^{(i)}|, |\mathcal{T}_i|\right)$$
$$+ 4\sqrt{\frac{d\log(2n) + \log\left(\frac{2}{\delta}\right)}{4n}} + \gamma, \tag{13}$$

as desired. □

## 2 COMPLETE EXPERIMENTAL RESULTS

The trends of accuracy and statistical parity among all six datasets are displayed as Figure 1 and Figure 2, respectively.

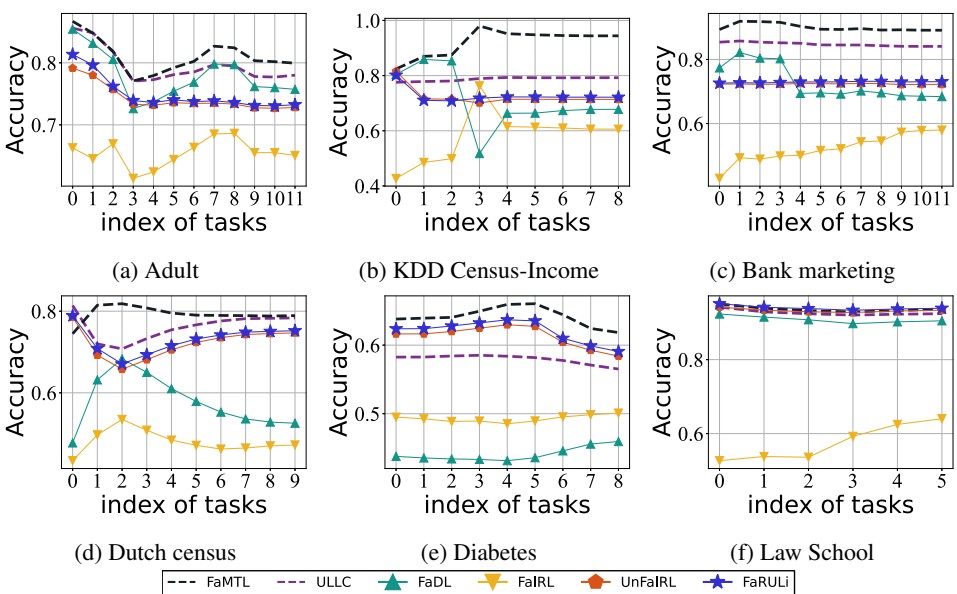

Figure 1: The trends of Accuracy of our FaRULi approach and its 5 competitors on 6 datasets.

## 3 RESEARCH QUESTION

We have discussed the improvement brought by the re-ordering function of our method. In the next research question, we will further analysis under what circumstances the re-ordering process will contribute the most, combing `Law School` dataset as an example.

**Q3.** *Under what circumstances will re-ordering lead to more substantial improvements?*

The introduction of re-ordering is meant to find the task that is closest to the current retained dataset among multiple tasks. Thus, when a newly observed task shows a distribution that is dissimilar to the current retained dataset, but tasks observed afterward exhibit similar distributions, the re-ordering can enhance the performance of our method. When comparing with the variant UnFaIRL, which is identical to FaRULi in all aspects except for the lack of the re-ordering function, the overall performance of FaRULiis 0.6% higher in accuracy and 2.4% lower in statistical parity than UnFaIRL among all six datasets. Especially for the `Law School` dataset, for example, FaRULi performs the best in both accuracy and statistical parity compared to all other methods observed from Table 1 (in main paper). Only as shown in the Figure 2f, FaRULi has the higher statistical parity values than FaMTL, which owns full label information, for the first two tasks. However, starting from $\mathcal{T}_2$, the statistical parity of FaRULi becomes similar to that of FaMTL, and the gap further widens by $\mathcal{T}_3$. This suggests that DT3 provides more fair knowledge to the retained dataset than $\mathcal{T}_1$. When

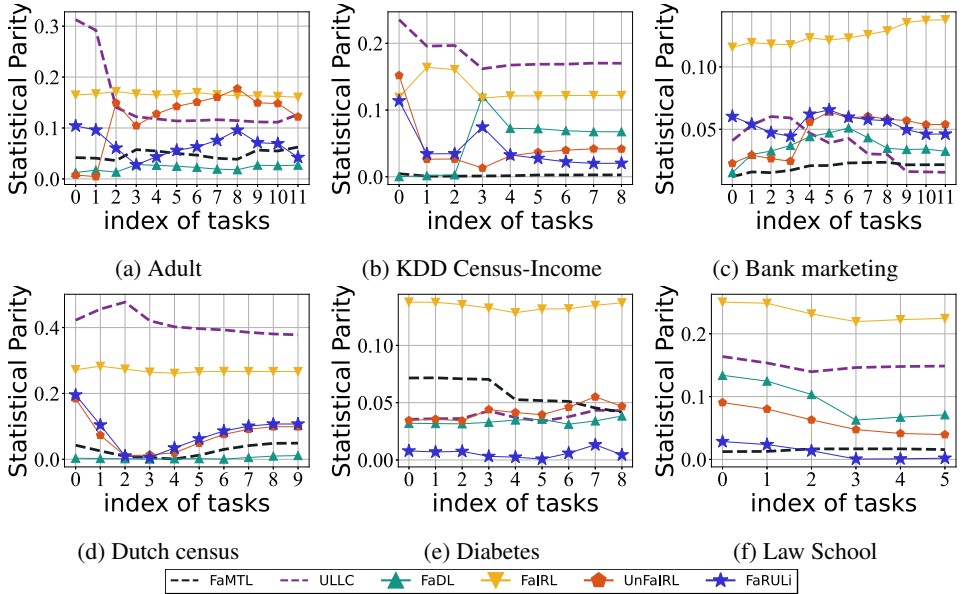

(a) Adult  (b) KDD Census-Income  (c) Bank marketing

(d) Dutch census  (e) Diabetes  (f) Law School

Figure 2: The trends of Statistical Parity of our FaRULi approach and its 5 competitors on 6 datasets.

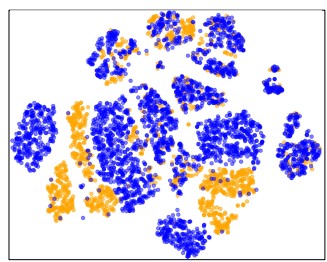

Figure 3: Distribution of `Law School` dataset. $\mathcal{T}_0$ and $\mathcal{T}_3$ are grouped, represented by yellow points; All other tasks are grouped, represented by blue points;

we reveal the learning order of FaRULi for `Law School`( $\mathcal{T}_0 \to \mathcal{T}_3 \to \mathcal{T}_2 \to \mathcal{T}_1 \to \mathcal{T}_5 \to \mathcal{T}_4$), we can observe that re-ordering indeed identified two tasks, $\mathcal{T}_3$ and $\mathcal{T}_2$, that are closer to $\mathcal{T}_0$ than the adjacent task $\mathcal{T}_1$, and subsequently altered the learning sequence, instead of using the default sequence: $\mathcal{T}_0 \to \mathcal{T}_1 \to \mathcal{T}_2 \to \mathcal{T}_3 \to \mathcal{T}_4 \to \mathcal{T}_5$. Moreover, when we consider $\mathcal{T}_0$ and $\mathcal{T}_3$ as one group and the other tasks as another group, we use the T-SNE (Van der Maaten & Hinton, 2008) to reduce instances of them into a two-dimensional space. As shown in Figure 3, the yellow points represent $\mathcal{T}_0$ and $\mathcal{T}_3$, and the blue points represent other tasks. Except for a few yellow points mixing with blue points, most yellow points in the two-dimensional space are concentrated in two distinct clusters from the blue points. This further substantiates that $\mathcal{T}_0$ and $\mathcal{T}_3$ constitute the most similar pair among all tasks. Hence, the four comparative methods that adopt lifelong learning but lack a re-ordering all yielded inferior results due to the wrong sequence. Even FaMTL, which possesses full label information and simultaneously learns all tasks, delivered slightly inferior results to our method due to the potential interference caused by the different distributions between the group of $\mathcal{T}_0$ and $\mathcal{T}_3$ and other tasks. Therefore, we can conclude that the introduction of the re-ordering is as we designed it to be: when the task adjacent to the retained dataset presents a distinct distribution, the re-ordering must alter the learning order to enhance the performance of our model.

# 4  IMPLEMENTATION DETAILS

## 4.1 Implementation Details

We implement FaRULi with PyTorch (Paszke et al., 2019) and all experiments are benchmarked on virtual machines, configured as 4 x Intel(R) Xeon(R) Gold 6148 CPU, one Nvidia V100 GPU, and 16GB RAM. The model of FaDL is implemented with the Fairlearn package (Bird et al., 2020).