# OpenReview forum: "Unsupervised Lifelong Learning with Sustained Representation Fairness"
_ICLR.cc/2024/Conference — Submitted to ICLR 2024_

### Official Review · Reviewer_an9h · 2023-10-19

**Soundness:** 3 good
**Presentation:** 3 good
**Contribution:** 3 good
**Rating:** 6
**Confidence:** 1

**Summary:**

This paper focuses on lifelong learning and its challenges in handling tasks that introduce false correlations between target labels and demographic attributes, resulting in bias. The authors claim that existing solutions often overlook the diversity of task distributions, particularly in lifelong learning, and struggle to maintain fairness. The paper introduces "Sustaining Fair Representations in Unsupervised Lifelong Learning" (FaRULi), inspired by human learning behavior, which prioritizes simpler tasks and adjusts task schedules based on fair representations.

**Strengths:**

The paper address a the problem of lifelong learning of fair DNN where new tasks come in resulting in an incremental improvement of the decision function. This is an interesting topic and needs to be studied from the fairness perspective.

**Weaknesses:**

Abstract: „Like human who tends to prioritize simpler tasks over more challenging ones that significantly outstrip one’s current knowledge scope, FaRULi reschedules a buffer of tasks based on the proximity of their fair representations.“: Is there any reference which claims that human tend to prioritize simpler tasks? Looks kind of obvious but there needs to be a scientific study which shows this.

The datasets used for evaluation are quite simple once’s - tabular data only. It’s shown that AdaBoost is why more accurate at tabular [1] data so in general why should one use DNN here?

Are there any kind of loopbacks studied? Like the algorithm does a decision which is used for new data which is used to further train the algorithm etc.?

[1] https://arxiv.org/abs/2106.03253

**Questions:**

- „in-domain bias and fail to generalize well to new tasks with data distributional shifts (Barrett et al.).“: Missing year in the reference
- 4.1. in the table you write „tasks“ what exactly are the tasks in the dataset? This is not really well explained in the text.
- There is a resent paper which uses stochastic quantized representations [2], this can ensures fair representation also after a distribution shift. However, I agree that the method proposed here is a more general way to address the problem of new incoming data.

[2] https://ojs.aaai.org/index.php/AAAI/article/view/25851

**Details Of Ethics Concerns:**

The paper proposes a new approach to have a life-long learning of fair algorithms which is needed when deploying such systems. A detailed legal analysis of the method is however still needed when deployed since it can directly effect discrimination.

---

> ### Author Response · Authors · 2023-11-20
> **The Response to Reviewer an9h**
>
> Thank you for your constructive comments. Below, we address your comments one by one in a Q&A fashion.
>
> Q1: Abstract: „Like human who tends to prioritize simpler tasks ...“: Is there any reference which claims that human tend to prioritize simpler tasks? Looks kind of obvious but there needs to be a scientific study which shows this.
>
> A1: This idea was firstly coined as 'starting small' in [1], stating that human beings can only learn complex and abstract concepts from assembling more intuitive understandings over simple objects. For example, students who want to learn calculus should start with basic arithmetic. This idea was later applied to the learning of neural networks by [2] and is known as curriculum learning.
>
> [1] Elman, Jeffrey L. "Learning and development in neural networks: The importance of starting small." *Cognition* 48, no. 1 (1993): 71-99.
>
> [2] Bengio, Yoshua, Jérôme Louradour, Ronan Collobert, and Jason Weston. "Curriculum learning." In *Proceedings of the 26th annual international conference on machine learning*, pp. 41-48. 2009.
>
> Q2: The datasets used for evaluation are quite simple once’s - tabular data only. It’s shown that AdaBoost is why more accurate at tabular data so in general why should one use DNN here?
>
> A2: We use an elastic DNN in this work because its depth can serve as an indirect metric to measure the distance between two tasks. We appreciate your suggestion and conduct experiments on the vision datasets. Please find out the answer in A2 of Reviewer 8gZZ due to character limitations.
>
> Q3: Are there any kind of loopbacks studied?
>
> A3: Our design will not incur loopback and was widely used in the literature, like [3]. This is because not all the predicted pseudo labels are integrated in the retained dataset; rather, only those predicted with high confidence and more likely to be correct labels are integrated. To bolster the confidence of these samples, we implemented a margin mechanism. More specifically, a classifier has been designed to determine the membership of each sample, distinguishing whether it originates from the learned retained dataset or a new task. Only those samples from new tasks that exhibit a probability exceeding 80 percent (a margin commonly used across most datasets) of belonging to the retained dataset, as determined by the classifier's sigmoid layer prediction, are actually added to the retained dataset
>
> [3] Sohn, Kihyuk, David Berthelot, Nicholas Carlini, Zizhao Zhang, Han Zhang, Colin A. Raffel, Ekin Dogus Cubuk, Alexey Kurakin, and Chun-Liang Li. "Fixmatch: Simplifying semi-supervised learning with consistency and confidence." *Advances in neural information processing systems* 33 (2020): 596-608.
>
> Q4: data distributional shifts (Barrett et al.).“: Missing year in the reference
> A4: We will correct this typo in camera ready.
>
> Q5: what exactly are the tasks in the dataset? This is not really well explained in the text.
>
> A5: Each task represents a distinct, non-overlapping subset of the original dataset. For instance, in the case of the adult dataset, individuals (instances) are categorized into 12 separate groups (tasks) based on their occupations, with examples including Tech-support, Sales, and Farming-fishing, among others. The rationale behind selecting occupation, as inspired by [4], is to ensure a varying distribution of P(Y|P) across different tasks. This approach leads to a dynamic proportion of males and females classified as having high incomes within various occupational categories. Once a model has effectively learned from a specific group of people (task), it is crucial to sustain its performance across other groups (tasks).
>
> [4] Le Quy, Tai, Arjun Roy, Vasileios Iosifidis, Wenbin Zhang, and Eirini Ntoutsi. "A survey on datasets for fairness‐aware machine learning." *Wiley Interdisciplinary Reviews: Data Mining and Knowledge Discovery* 12, no. 3 (2022): e1452.
>
> Q6: There is a resent paper which uses stochastic quantized representations [2], this can ensures fair representation also after a distribution shift.
>
> A6: Thanks for bringing this paper into our radar. We scrutinized the paper and found that it focused on a different learning problem from ours. Specifically, the paper [5] proposed an alternative de-biasing function through the random (stochastic) projection of discriminative feature representations. However, there is no evidence to show that such a de-biasing function can hold invariant across different tasks, each of which has a distinct demographic distribution. Thus, their proposed method cannot be directly adapted into our lifelong learning context. We will add this discussion in our camera ready.
>
> [5] Cerrato, Mattia, Marius Köppel, Roberto Esposito, and Stefan Kramer. "Invariant representations with stochastically quantized neural networks." In *Proceedings of the AAAI Conference on Artificial Intelligence*, vol. 37, no. 6, pp. 6962-6970. 2023.

---

### Official Review · Reviewer_kvtG · 2023-10-22

**Soundness:** 2 fair
**Presentation:** 2 fair
**Contribution:** 2 fair
**Rating:** 3
**Confidence:** 4

**Summary:**

The paper tackles the problem of sustaining fairness in decisions made in an incremental learning setting. To that end, the paper introduces an approach that relies on task labels of only the first subtask, and other tasks can be unlabeled (for the target) attribute. The paper relies on an adversarial objective to learn fair representations that tries to maximize the utility of the learned representations while suppressing the protected attribute and task identity. The paper introduces a method involving elastic networks and applies the adversarial objective at every layer of the network. The proposed approach also reorders the incoming task to ensure similar tasks are trained together. The paper also provides a theoretical analysis of their approach and showcases results on tabular datasets.

**Strengths:**

1. The proposed approach, FaRULi, achieves significant improvement over baseline approaches on several tabular datasets.
2. The paper provides a theoretical study of their approach.

**Weaknesses:**

1. The primary weakness of the paper is that it is not well-written. Several parts of the paper are very hard to follow (see comments below). Much of the confusion stems from the fact that the paper does not introduce elastic networks and their functioning before the method section.
2. The technical contribution is limited given the fact FaRULi is a simple application of an adversarial objective in every layer of the network. This adversarial objective is well studied in the literature except for the task identification term, which is not well motivated in the paper.
3. The paper needs more justification to support/clarify why their approach works so well. The theoretical section provides some insights into it but would be interesting to see how these theoretical results compare with a simple baseline, e.g. an adversarial network.
4. The experimental section is limited as the paper provides results on tabular datasets only. More experiments on complex data like vision and NLP datasets can be performed to evaluate the efficacy of FaRULi (the FaIRL paper has a similar set of experiments).
5. The claim of being unsupervised is not well supported as the method uses task labels for the first task. Moreover, since the focus is on learning fair representations the protected label should ideally be considered as the primary supervision. On a similar note, comparison with some of the baselines may be slightly unfair because they do not rely on task labels.


Comments:

Introduction, para - 3: "This strategy ..." - this sentence is quite hard to understand.

Specific contributions: (i) "fairness" -> fair

Problem statement: may use a different variable than P_i for the protected feature as P is also used for probability

Sentence before Eq. 2: what does "norm"  mean here?

Next para: "elastic network that ..." -> "elastic network such that ..."

Section 2.1: 2nd line: "eliminate" -> eliminating

Section 2.1: 13th line: what does "both tasks" refer to?

Eq. 3: it is not clear to me why similarity between representations of R_0 and T_i is desired.

Section 2.2 1st sentence: terms like "propagating fair knowledge" are unclear to me. Please make this sentence a bit smoother.

Same paragraph: it is very difficult to understand the functioning of EFRL. The full form of this term was provided in the introduction only without any citations. It would help to have a section on it or describe it using citations of previous works.

There are several terms in the same paragraph like "negative model use", "independent classifier groups" which are unclear to me. They need to be defined earlier on in the paper.

Sentence after Eq. 4: "i-th layer" -> "l-th layer"?

The equation of a^{(l)} is very hard to understand. It took me some time to get that \sum L_EFRL is a power of \beta.

Section 3, unclear what "detriment learning performance" means in this context?

**Questions:**

1. The setup is unclear to me. If FaRULi performs a reordering of the learning tasks after it receives different tasks when exactly are the predictions of the task recorded? Is this consistent with the baseline approaches?
2. Section 2.3, it is unclear to me how Q measures the distance between different tasks?
3. Theorem 1, where is the loss L_FaRULi defined?

---

> ### Author Response · Authors · 2023-11-20
> **The Response to Reviewer kvtG**
>
> Thank you for your constructive comments. Below, we address your comments one by one in a Q&A fashion.
>
> Q1:  The elastic network design .., making the EFRL ..., why not have a section ...?
>
> A1: We respectfully point out that the elastic network and EFRL are the same, detailed in the Introduction (Page 2) and Section 2.2 (Page 3) with intuition and technical specifics, respectively. The presentation follows your suggested structure.
>
> Q2: The task identification term ..., Why similarity between ..., unclear what "detriment ..." means?
>
> A2: We respectfully point out that our work focuses on lifelong adaptation of fair representations to unlabeled tasks, not just single-task learning via an adversarial network. Challenges stem from unseen negative adaptation, owing to a lack of labels and demographic shifts, potentially leading to error build-up. The EFRL network addresses this by using depth information to select $\mathcal{T}_i$, the task with a distribution closest to $R^{(0)}$. '...detriment...' implies that choosing a suboptimal model depth can lead to negative model transfer between tasks, adversely affecting the fair representation learning process.
>
> [1] https://proceedings.mlr.press/v37/ganin15.html
>
> [2] https://openaccess.thecvf.com/content_cvpr_2018/html/Pinheiro_Unsupervised_Domain_Adaptation_CVPR_2018_paper.html
>
> Q3: It would be interesting to see how ....
>
> A3: Theorem 1 indicates our algorithm's performance is similar to an optimal-depth adversarial network, which is infeasible in practice, with comparable cumulative loss bounded by $O(\frac{\ln(L)}{1-\beta})$.
> FaDL serves as the baseline and our results show that FaDL fails to generalize fair representations across different tasks, underperforming our algorithm by 7.6% in accuracy on average.
>
> Q4: "This strategy ..." is quite hard to understand.
>
> A4: Divergent demographic distributions means that P (Y | P) varies across different tasks at various extends. We display this for the Adult dataset only in the table below due to the character limitations.
> "Incrementing fair representation" extracts latent features for learned data $R^{(i-1)}$ and new task $\mathcal{T}_i$. This process is efficient due to a similar demographic distribution between $R^{(i-1)}$ and $\mathcal{T}_i$, achieved by starting with similar tasks and leaving distinct tasks for later. Gradually, $R^{(i-1)}$ incorporates various aspects of prior tasks, aligning with diverse task distributions. The feature extractor's parameters are reused, enhancing the incremental nature of the representation process.
>
> Distribution of P(Y=1|P=1) / P(Y=1|P=0) across all tasks in Adult.
> |Dataset|T_1|T_2|T_3|T_4|T_5|T_6|T_7|T_8|T_9|T_10|T_11|T_12|
> |-------|---|---|---|---|---|---|---|---|---|----|----|----|
> |Adult|0.624/0.376|0.842/0.158|0.971/0.029|0.849/0.151|0.982/0.018|0.923/0.077|0.933/0.067|0.609/0.391|0.975/0.025|0.785/0.215|0.948/0.052|0.907/0.093|
>
> Q5: The paper provides results on tabular datasets only.
>
> A5: Please find the answer in A2 of Reviewer 8gZZ due to character limitations.
>
> Q6: The claim of being unsupervised..., On a similar note, some baselines may do not rely on task labels.
>
> A6: We agree that coining our regime as purely “unsupervised” is not rigor. However, given its reliance on labeling a single task in a multi task context, it is less burdensome than semi-supervised approaches requiring labels for all tasks,
> distinguishing our approach from those like FaDL and FaIRL, which necessitate full label information.
>
> Q7: When exactly are the predictions ...? Is this consistent ...? How Q measures ...?
>
> A7: In each round, $\mathcal{T}_i$ is chosen for prediction based on similarity to $R^{(i-1)}$. The process stops when adversarial training converges, marked by reaching a maximum number of epochs. The task with the lowest Q value is then chosen.
> All competing models follow the same setup. Tasks are split into training and testing subsets where testing subsets are used to calculate accuracy and statistical parity after learning on training subsets. A high Q suggests either uniform weight distribution across layers or significant weights in deeper layers, implying a deep network. Conversely, a low Q indicates that shallow layers are more important.
>
> Q8: Where is the loss L_FaRULi defined?
>
> A8: $\mathcal{L}_{FaRULi}$ is the cumulative loss of all layers over T rounds. Details can be found in Appendix 1.1.
>
> Q9: "negative model use" need to be defined earlier on in the paper
>
> A9: The term is first introduced on Page 2 (Introduction) with citing papers. It's akin to "negative transfer" [3], which means a large difference between learned and new tasks makes reusing the trained model be less effective than starting with a new model for the new task.
>
> [3]https://openaccess.thecvf.com/content_CVPR_2019/html/Wang_Characterizing_and_Avoiding_Negative_Transfer_CVPR_2019_paper.html
>
> Final, we appreciate your suggestions and will refine our language and correct typos for better clarity.

---

> ### Author Response · Authors · 2023-11-21
> **Rebuttal follow-up**
>
> Dear Reviewer,
>
> We wanted to check if you had the opportunity to review our rebuttal and the revised manuscript (rebuttal revision) for your reference. We are committed to addressing any concerns or questions you may have about our work. If there are any aspects of the rebuttal or the manuscript that require further clarification, or if you have any new questions, please do not hesitate to let us know. We are eager to provide any additional information that might help in your evaluation of our paper.
>
> Thank you once again for your valuable insights and we look forward to your response.
>
> Best,
> Authors

---

### Official Review · Reviewer_8gZZ · 2023-10-25

**Soundness:** 3 good
**Presentation:** 2 fair
**Contribution:** 2 fair
**Rating:** 5
**Confidence:** 2

**Summary:**

This paper discusses the challenge of lifelong learning to improve decision-making while ensuring fairness. The authors introduce a new approach called FaRULi, which prioritizes simpler tasks over challenging ones and reschedules tasks based on their fair representations, starting with similar ones to accumulate de-biasing knowledge. It shows promise in making fair and accurate decisions in a sequence of tasks without supervision labels, backed by theory and empirical evaluation.

**Strengths:**

- This paper addresses the existing limitations in fairness representation learning by adopting a lifelong perspective, wherein it establishes a common representation space across a series of unlabeled tasks.

- The suggested approach appears innovative, and the theoretical framework appears well-founded.

- The method's overview, as illustrated in Figure 1, is presented in a clear and comprehensible manner to showcase its functionality.

- The theoretical analysis section is robust in substantiating the approach.

**Weaknesses:**

- The information in Table 1 is presented in a manner that makes it challenging to interpret. To facilitate comparisons, it is advisable to emphasize the highest accuracy and the lowest fairness violation. In light of the results, it's apparent that FaRULi only manages to achieve a somewhat modest level of performance concerning accuracy and fairness. This raises the question of how this outcome supports the claim that "On average, our model surpasses three leading competitors by 12.7% in prediction accuracy and by a substantial 42.8% in terms of ensuring statistical parity."

- The experiment exclusively utilizes tabular data. However, it raises the question of how this proposed approach can effectively address more intricate tasks, such as those in the domains of computer vision or natural language processing.

**Questions:**

This paper effectively addresses a significant practical concern related to lifelong learning. However, the experimental section may not be entirely persuasive. Despite reading section Q1 on how FaRULi works, I remain somewhat perplexed. The process for making comparisons doesn't appear straightforward. As I noted in the weaknesses, the results appear to hover around the threshold, without demonstrating clear superiority over other baseline methods.

---

> ### Author Response · Authors · 2023-11-20
> **The Response to Reviewer 8gZZ**
>
> Thank you for your constructive comments. Below, we address your comments one by one in a Q&A fashion.
>
> Q1: Are the statistics calculated from Table 1 correct? I fail to map why it is the case that "On average, our model surpasses three leading competitors by 12.7% in prediction accuracy and by a substantial 42.8% in terms of ensuring statistical parity"?
>
> A1: We respectfully point out that only FaDL and FaIRL are our rival models, representing the state of the art in the FRL context. Compared to FaDL and FaIRL, our proposed approach demonstrated significant performance improvement in terms of prediction accuracy by 20.4% and statistical parity by 44.4%. The other methods including UnFaIRL, FaMTL, and ULLC are not direct competitors. Specifically, UnFaIRL without optimizing the learning order is the variant of our proposed approach, crafted for ablation study. FaMTL and ULLC represent the performance upper bound in our context. Namely, FaMTL learns all tasks with all labels at once, where the fairness constraint is enforced in such offline multi-task learning setting, thus it always presents the best accuracy performance. While ULLC can learn tasks in sequence with only one labeled task (like our algorithm does), it does not impose fairness constraints. We thus can observe that ULLC slightly outperforms our algorithm in accuracy, but is inferior to ours in the fairness metric by substantial margins.
>
>
> Q2: Can the proposed algorithm be generalized to high-dimensional data like images in addition to the studied tabular data?
>
> A2: Yes, we followed [1] to construct an MNIST image dataset, with the protected feature being the background color of each digit type. Specifically, MNIST is divided into five tasks, each containing ten digits from 0 to 9. These tasks differ in two aspects. First, the class distribution P(Y) varies across tasks, with the largest number of images representing a certain digit type differing from one task to the others. Second, we designated red and green colors as the protected features, and the conditional P(Y|P) also varies. For instance, Task 1 may have more digit "1" in red and digit "9" in green, while the other tasks may exhibit reversed patterns. Here, we intentionally designated Task 3 ($\mathcal{T}_3$) as the most distant task, in which its conditional P(Y|P) differs significantly from all the other four tasks. Intuitively, a fair classifier should not establish a superficial correlation between color and digit, thus accurately predicting an input image regardless of its background color. We compared our FaRULi model with FaDL, and the results are presented in the table below. $\mathcal{T}_1$ is selected as the initial task with full labels. From the results, we observed that our model outperforms FaDL by 3.4% and 0.148 in Accuracy and Equalized odds (EO) on average, respectively. Moreover, FaDL performs better in the first two tasks $\mathcal{T}_1$ and $\mathcal{T}_2$ (average accuracy = 92.5%, average EO = 0.242) but cannot generalize to other tasks. We extrapolate that the function to debias color from digit type learned in $\mathcal{T}_1$ and $\mathcal{T}_2$ by FaDL cannot be adapted to other tasks, even if $\mathcal{T}_3$ has slightly different P(Y|P). In contrast, our model maintains good performance on $\mathcal{T}_1$, $\mathcal{T}_2$, $\mathcal{T}_4$, and $\mathcal{T}_5$, all of which have different conditional P(Y|P). On the most distant $\mathcal{T}_3$, our model achieves comparable accuracy performance as FaDL. This aligns with our assumption that the model cannot generalize to a highly dissimilar task, and our model resolves the negative transfer issue by prioritizing $\mathcal{T}_2$, $\mathcal{T}_4$, and $\mathcal{T}_5$ and achieving significantly better EO performance on them compared to FaDL.
>
> | MNIST           | Task-1 | Task-2 | Task-3 | Task-4 | Task-5 |
> |-------------------|------------|--------|--------|--------|--------|
> | FaDL Accuracy     | 0.939      | 0.911  | 0.855  | 0.851  | 0.886  |
> | FaDL EO           | 0.267      | 0.217  | 0.396  | 0.379  | 0.334  |
> | Our Model Accuracy| 0.952      | 0.929  | 0.887  | 0.946  | 0.933  |
> | Our Model EO      | 0.093      | 0.164  | 0.218  | 0.099  | 0.130  |
>
> [1] Bahng, Hyojin, Sanghyuk Chun, Sangdoo Yun, Jaegul Choo, and Seong Joon Oh. "Learning de-biased representations with biased representations." In *International Conference on Machine Learning*, pp. 528-539. PMLR, 2020.

---

> ### Author Response · Authors · 2023-11-21
> **Rebuttal follow-up**
>
> Dear Reviewer,
>
> We wanted to check if you had the opportunity to review our rebuttal and the revised manuscript (rebuttal revision) for your reference. We are committed to addressing any concerns or questions you may have about our work. If there are any aspects of the rebuttal or the manuscript that require further clarification, or if you have any new questions, please do not hesitate to let us know. We are eager to provide any additional information that might help in your evaluation of our paper.
>
> Thank you once again for your valuable insights and we look forward to your response.
>
> Best,
> Authors

---

### Official Review · Reviewer_E5F7 · 2023-10-26

**Soundness:** 2 fair
**Presentation:** 3 good
**Contribution:** 2 fair
**Rating:** 5
**Confidence:** 3

**Summary:**

The authors propose a new unsupervised lifelong learning scheme incorporating a statistical parity fairness constraint. The method broadly works as follows:
1) Train a model on the initial task, for which target labels are assumed to be available.
2) Consider two new candidate tasks for which labels are unavailable. For each of them, perform unsupervised domain adaptation to adapt the model's latent representation (that is shared across tasks) such that the observed distributions in both tasks are aligned. The task for which this can be achieved by a shallower model is then selected as the next task, the rationale being that this should be a more similar task that induces a lower risk of catastrophic forgetting.
3) For the selected task, generate pseudolabels for high-confidence predictions made by the model, and incorporate these into the retained training set for the following tasks to come.
4) Repeat this process for all new incoming (unlabeled) tasks.

In each of these steps, a statistical parity loss component is taken into account, which penalizes the model for making positive predictions more often for one group of input data compared to another. (The protected attribute is assumed to be observed.) This corresponds to lifelong unsupervised fair representation learning, or to the notion of marginal invariance in the domain adaptation literature.

In a series of experiments on standard tabular datasets, the proposed method compares favorably against a range of different baselines and competitor methods, both in terms of overall prediction accuracy and statistical parity.

Finally, the authors also provide theoretical bounds on the performance of their method.

**Strengths:**

The authors address an important, urgent, and essentially unsolved problem: maintaining the fairness of a model while adapting to new data in an unsupervised setting. I have not seen much prior work in this area, and any progress on this topic is highly welcome.

The framework proposed by the authors is relatively general and should be easy to adapt to other settings and, in particular, other fairness constraints. Empirically, it appears to perform well.

Methodologically, the selection of the next task to adapt to based on the depth of the required changes to the model's representation seems novel to me, although I am not an expert in lifelong learning and hence cannot be certain about this.

**Weaknesses:**

I see three main weaknesses of the manuscript in its current form, all of which I believe can be addressed with reasonable effort.

### **1. The choice, enforcement, and evaluation of the fairness constraint**
The authors choose demographic/statistical parity as their fairness criterion of choice, which asks that subjects from group A receive positive predictions with equal likelihood as those from group B *regardless of the two groups' baseline likelihoods of receiving a positive outcome*. For example, in the diabetes task, this would require that even if subjects from group A have a much higher prevalence of diabetes, the model would still have to predict diabetes at equal rates for both groups, essentially forcing the model to make false predictions. There are scenarios where this may be considered a desirable outcome, but they seem rather rare to me. For this reason, equal opportunity or equalized odds are much more widely used criteria in the algorithmic fairness literature. (In their discussion of fairness definitions, the authors completely gloss over these two absolute standard criteria.)

Like discussed above, enforcing statistical parity in the face of baseline/prevalence differences forces model misclassifications. This is likely to have strongly affected the results obtained by the authors in the different tasks - they may simply be varying in their level of baseline differences. Moreover, the degree to which this will affect model performance/accuracy will differ based on how strongly statistical parity is enforced / the choice of the regularization constant, which seems likely to differ between the compared methods. These two factors may have strongly influenced the results obtained by the authors, maybe even more so than any intrinsic differences between the datasets/tasks and modeling/training approaches. (This might explain, for example, why FaDL obtains worse accuracy but better statistical parity on the Dutch dataset, as noted by the authors: it simply was more "successful" at enforcing statistical parity, which is fundamentally incompatible with high accuracy in datasets with strong baserate differences.)

In this regard, I would suggest:
- Implementing at least one other kind of fairness constraint, preferably equal opportunity or equalized odds. I believe that due to the generality of the approach presented by the authors, this should be feasible with reasonable effort? I would expect much less of a fairness-performance trade-off with these constraints, especially equal opportunity.
- Reporting the baseline incidences $P(Y \mid P)$ for all protected groups in all datasets.
- Adding single-task supervised unconstrained and fairness-constrained baselines for each of the datasets, in order to get a better sense of upper performance bounds.
- Assessing the impact of the fairness regularization constants, or tuning these such that all approaches achieve comparable levels of fairness/unfairness.

Some essential references on these issues:
- Hardt et al., Equality of Opportunity in Supervised Learning, https://arxiv.org/pdf/1610.02413.pdf
- https://fairmlbook.org/classification.html
- Zhao and Gordon, Inherent Tradeoffs in Learning Fair Representations, https://jmlr.org/papers/v23/21-1427.html
- Mehrabi et al., A Survey on Bias and Fairness in Machine Learning, https://dl.acm.org/doi/10.1145/3457607
- Zhang et al., Improving the fairness of chest x-ray classifiers, https://proceedings.mlr.press/v174/zhang22a, is one of many prior works that study the influence of the regularization constant in fairness-constrained learning approaches.


### **2. Language issues**
While the manuscript is organized well overall and was mostly understandable, there are quite a few places where I stumbled upon sentences or formulations that did not seem to make any sense to me. I will remark on a some specific instances below, but I suggest that the authors carefully revise their language for precision and clarity throughout the manuscript.


### **3. Unproven claims concerning the task selection mechanism**
In the last paragraph on page 4, and a few other places, the authors present a long list of conjectures on what their method will probably do: "deep layers will outperform... weight parameters of deep layers [will] start to exceed that of shallow layers ... the parameters of shallow layers [will] initially increase sharply ... weights of the deepest layers [will] increase last and remain relatively small ..."

All of these seem to me to just be conjectures, however: none of these claims are proven experimentally or theoretically. I would suggest either making clear that these are pure hypotheses, or demonstrating these effects experimentally.

Moreover, this is just one (new) method to assess task similarity / distributional differences between successive tasks. Could the authors either compare to one or two other standard distributional similarity metrics (for choosing which task to incorporate next), or provide some kind of argument for why existing similarity metrics are not applicable here?

**Questions:**

I will use this space for minor remarks.
- In the introduction on p.2, what does "more divergent demographic distributions" mean, and what does "incrementing a fair representation" mean? This whole paragraph was not very clear to me; at the end of the introduction I had not really understood what the authors set out to do.
- On p.3, I suggest revising "our FaRULi approach takes a norm as follows" and the sentence "Disparate distribution between incoming and learned tasks invokes a deep model, indicating the occurrence of negative model reuse." (More generally, "disparate distribution" is not a usual term; I suggest rephrasing this to more precisely described what the authors want to say.)
- On p.3 towards the bottom, the authors write that "..., a comparable performance on Ti can be achieved" which seems like an overly optimistic claim.
- Also on the bottom of p.3, the authors write that "... disparate distributions [will] likely lead to overfitting in R(0) and yield unpredictable, even negative performance on Ti." This is, again, just an unproven hypothesis, I believe? I also suggest revising the sentence "We draw inspirations from that approximating two disparate distributions necessitates a more complex model."
- In the caption of Fig. 1 on p.4, I suggest revising the sentence "R(0) is initialed and engaged in fair representation learning [...]."
- On p.5, I suggest revising the sentences "As disparate distributions invoking the deep network, we introduce a similarity-measurement metric [...]." and "Therefore, with the help of re-ordering, instances from T2 but being misclassified into R(0) via the red circle and their high confidence pseudo labels [...] are retained and incorporated into the retained dataset [...]." (I have no idea what this sentence is supposed to mean.)
- On p.8, I do not understand the sentence "To ensure fairness, both competitors utilize pseudo-labels for replay to ensure the lifelong learning." - what do the replayed pseudo-labels have to do with ensuring fairness?
- On p.9, the authors write "Bias in ML models can be attributed to the information carried by protected features." This is a highly simplistic characterization of bias. There are many different forms of bias that have many different causes; see, e.g., the Mehrabi survey I linked to above.
- A little further below (but in the same paragraph), the authors write that "While these frameworks demonstrate promising results in a specific domain, they fall short in maintaining fairness when applied to o.o.d. data (Barrett et al.)." The sentence before this one is about counterfactual fairness, suggesting that Barrett et al. show that counterfactual fairness "falls short in maintaining fairness when applied to o.o.d. data". They do not, however; in fact, Barrett et al. do not consider counterfactual fairness at all.

---

> ### Author Response · Authors · 2023-11-20
> **The Response to Reviewer E5F7**
>
> Thank you for your constructive comments. Below, we address your comments one by one in a Q&A fashion.
> Due to characters limitations, we are only reporting parts of the experimental results in your questions.
>
> Q1: Can the fairness constraint ..., such as equalized odds (EO)?
>
> A1: Yes, our proposed framework is general, allowing for various fairness metrics. Based on your suggestion, we've implemented EO, presenting parts of results as follows:
>
> | Dataset  | FaDL  | FaIRL | FaRULi |
> |-----|-----|------|-------|
> | Diabetes | 0.068 | 0.051 | 0.031  |
> | Law      | 0.234 | 0.091 | 0.087  |
>
> We can observe that the empirical superiority of our proposed FaRULi algorithm is invariant to the fairness metric, and our algorithm outperforms its two competitors FaDL and FaIRL in both datasets with 0.031 and 0.087, respectively.
>
> Q2: Reporting P(Y|P). Please clarify what "more divergent ..." and "incrementing ..." mean?
>
> A2: Due to character limitations,  please find our answers in A4 of Reviewer kvtG.
>
> Q3: Types of bias in ML models,
>
> A3: Yes, we're aware of the survey and have cited it. Our focus is on "fair representation learning (FRL)," not machine learning bias. Existing FRL studies, like [1], mainly address deriving fair data representations from unfair data. We'll emphasize our sub-domain and differentiate it from other biases in camera-ready.
>
> [1] https://dl.acm.org/doi/abs/10.1145/3278721.3278779
>
> Q4: Supervised with or w/o fairness-constrained baselines.
>
> A4: Results of Adult are shown as below.
>
> |Constraint|T_1|T_2|T_3|T_4|T_5|T_6|T_7|T_8|T_9|T_10|T_11|T_12|
> |----------|---|---|---|---|---|---|---|---|---|----|----|----|
> |w/o Acc|0.872|0.77|0.79|0.76|0.859|0.909|0.844|0.951|0.755|0.751|0.726|0.821|
> |w/o EO|0.101|0.173|0.167|0.138|0.643|0.688|0.286|0.044|0.216|0.221|0.278|0.195|
> |with Acc|0.817|0.665|0.775|0.566|0.857|0.856|0.842|0.932|0.761|0.589|0.687|0.715|
> |with EO|0.082|0.152|0.049|0.219|0.039|0.167|0.071|0.24|0.055|0.088|0.47|0.141|
>
> Q5: Fairness regularization constants
>
> A5: Yes, our experiments on the Law School dataset show that increasing $\lambda_1$ decreased accuracy from 91.6% to 4.7%, demonstrating that strict fairness constraints can lower accuracy and affect fairness. Both statistical parity (SP) and EO values fall to zero, with the learner classifying all instances as Y=0 to avoid unfair outcomes.
>
> | $\lambda_1$ = |0.01|0.03|0.05|0.1|0.5|1|
> |------------|-------|-------|-------|-------|-------|-------|
> | Acc| 0.916 |0.939|0.899|0.941|0.047|0.047|
> | SP|0.162|0.002 |0.054 |0.012 |0|0|
> | EO|0.269|0.087|0.107|0.050|0|0|
>
> Q6: The dynamics of ... be conjectures.
>
> A6: We respectfully disagree, referencing studies in traditional BP [2-3] and adversarial learning [4]. These show that while deeper neural networks offer more representation capacity, they're also more sensitive to initializations, potentially losing correlation with shallow features. Our backbone architecture is built upon [4], and our EFRL network design can enjoy the same merit as the elastic network in [4] does. We will further add experimental results as [3] did to support the intuition behind our design.
>
> [2] https://arxiv.org/abs/1605.07648
>
> [3] https://link.springer.com/chapter/10.1007/978-3-319-46493-0_39
>
> [4] https://dl.acm.org/doi/abs/10.1145/3442381.3449839
>
> Q7: Please clarify the correctness ...
>
> A7: Due to the character limitations,  please find our answers in A9 of Reviewer kvtG.
>
> Q8: Could the authors either compare ..., or provide some ...here?
>
> A8: We argue that traditional distance metrics are unsuitable due to two reasons: they struggle with high-dimensional data [5] and require predefined thresholds to distinguish "close" from "faraway" tasks, which aren't feasible in lifelong learning. Our experiments on high-dimensional image datasets are detailed in A2 of Reviewer 8gZZ.
>
> [5] https://kops.uni-konstanz.de/entities/publication/7cac4320-01b6-4403-b38a-ba87d550bc0a
>
> Q9: "..., a comparable performance ..." which seems like...
>
> A9: We'll revise our statement: If a shared subspace between $\mathcal{T}_i$ and $R^{(0)}$ is found, generalization from fully labeled $R^{(0)}$ to unlabeled $\mathcal{T}_i$ is possible, proved by many UDA studies [6-7]. However, such $\mathcal{T}_i$ may not always be feasible due to significant task differences or starting with an outlier task. Therefore, we will adjust our wording to "comparable performance on $\mathcal{T}_i$ is more likely" to cover these possibilities.
>
> [6] https://proceedings.mlr.press/v37/ganin15.html
>
> [7] https://openaccess.thecvf.com/content_cvpr_2018/html/Pinheiro_Unsupervised_Domain_Adaptation_CVPR_2018_paper.html
>
> Q10: how the replayed pseudo-labels ensure fairness?
>
> A10: We apologize for any confusion. Here, "fairness" means setting up a fair experimental benchmark, not fairness in machine learning. It's about equal conditions for all compared methods, which we'll clarify in camera-ready.
>
> Final, we will streamline our language for ease of understanding.

---

> > ### Comment · Reviewer_E5F7 · 2023-11-20
> >
> > Thank you for the thorough response and the additional experiments! Will you upload a revised manuscript (with changes highlighted) so that I and the other reviewers can look at the changes in detail? (Also, you can simply post multiple separate comments / answers to get around the character limit.)

---

> > > ### Author Response · Authors · 2023-11-21
> > > **The Response to Reviewer E5F7**
> > >
> > > Thank you for your suggestion. We have uploaded the rebuttal revision,
> > > in which our rectifications are highlighted in blue for better visibility.

---

> > > > ### Comment · Reviewer_E5F7 · 2023-11-22
> > > >
> > > > I would like to thank the authors for their quite substantial revision, including a lot of additional experiments, and the rebuttal(s)!
> > > >
> > > > I appreciate that several of my (and the other reviewers') concerns have been considered, including the addition of new experiments concerning the effect of the regularization constant , an extension to a simple image classification scenario, and the additional empirical evidence concerning the internal behavior of the EFRL network.
> > > >
> > > > However, several issues still remain (and for this reason I am not changing my score):
> > > > - While I strongly appreciate the addition of the new Figure 3, it seems to me that it actually does not show what the caption claims it shows? Maybe I am missing something, but the blue+red bars all seem to sum to 1, indicating that the bars in fact show something like P(P|Y=1), and not P(Y=1|P) as claimed?
> > > > - Concerning the issue of enforcing equalized odds instead of statistical parity, it currently reads as though the authors still enforce statistical parity and just _evaluate_ on equalized odds? Specifically, L_Obfs is designed "to make g(·) fail to infer the demographic group, thus obfuscate protected information from data representations from both tasks" - which, again, enforces SP and will hugely negatively impact model performance (and not do anything for fairness) in the case of baseline differences. (Also, the EO equation at the top of p.2 is wrong / ill-defined. Should be something like EO = |P(Yhat=1 | P=0, Y=0) - P(Yhat=1 | P=1, Y=0)| + |P(Yhat=1 | P=0, Y=1) - P(Yhat=1 | P=1, Y=0)|.)
> > > > - It is currently unclear to me how the baseline experiments shown in Fig. 7 were performed. Was this also a constrained-optimization approach? And if yes, which of the two discussed fairness constraints (EO, SP) were enforced here? Ideally, these numbers should also be shown in Fig. 6 to make it easier to compare the performance of the other methods to these results.

---

> > > > > ### Author Response · Authors · 2023-11-22
> > > > > **Response to Reviewer E5F7**
> > > > >
> > > > > We are grateful for your very constructive comments and would like to clarify them as follows.
> > > > >
> > > > > 1. We acknowledge that the presentation in the original Figure 3 may have led to some confusion. The intention behind the figure was to illustrate the demographic distribution shift across tasks by normalizing the conditional probabilities P(Y=1 | P=0) and P(Y=1 | P=1) to sum up to 1. This was done to enable a clearer visual comparison of the proportions of each demographic group within each task.
> > > > > However, based on your feedback, we realize that this normalization may have obscured the shift of class distributions. To rectify this, we have updated Figure 3 in the revised manuscript. The normalization has been removed, and the combined length of the red and blue bars representing P(Y=1) in its true proportion now reflect the class (granted) distribution of each task. We agree that this new Figure 3 allows for a more direct and accurate interpretation of both demographic and class distribution shifts across tasks. We have revised this in the new rebuttal version.
> > > > >
> > > > > 2. We would like to further clarify our stance from two aspects. First, our adversarial learning mechanism, implemented through L_Obfs, does not explicitly enforce SP or EO. Instead, the objective is to obfuscate protected feature information within the resultant latent representations. We envision that This approach indirectly contributes to improved SP and EO performance, and has been supported by several prior studies [1, 2] in which adversarial learning similar to ours have been implemented effectively. Furthermore, other methods such as those employing mutual information [3, 4] and distance measurement [5, 6] have also been used in similar contexts, without direct modeling of SP or EO. Our approach aligns with these studies, using SP and EO as evaluation metrics to tune our model on validation data and report empirical results on test data.
> > > > >
> > > > > Second, regarding the EO definition, we opted for a different formulation: EO = max { |P(Yhat=1 | P=0, Y) - P(Yhat=1 | P=1, Y)| }. Although different from your suggested equation, both formulations are valid. The original definition of EO, as found in [7], is expressed as an equality constraint: P(Yhat=1 | P=0, Y) = P(Yhat=1 | P=1, Y), for all Y={+1, -1}. In our case, using the maximum over all classes Y, rather than a summation, was a deliberate choice to avoid inflated EO values, particularly when dealing with datasets like Bias-MNIST which has multiple (ten) classes. This approach facilitates clearer comparisons across different datasets. The use of the maximum in EO calculations is also a practice observed in other works [7, 8].We have rectified our EO definition on page 2.
> > > > >
> > > > > 3. The baseline models in Figure 7 are independent classifiers, each trained separately on their respective tasks with full label access. When fairness constraints are considered, these models transform into FaDL classifiers which use adversarial learning for the debiasing of the protected feature P, aligning with our implementation. Note, these baseline results are not directly comparable with those presented in Figure 6, as the SP and EO in Figure 7 are computed individually for each task. In contrast, the metrics in Figure 6 are cumulative, representing an aggregation of results from all preceding tasks. For instance, the EO metric at Task T_4 in Figure 6 is an average of the outcomes from Tasks T_0 through T_4. This approach is intentional, as Figure 6 aims to demonstrate the efficacy of our proposed method in a lifelong learning scenario. Our algorithm showcases robust performance across tasks, maintaining high accuracy while keeping SP and EO metrics low. This is in contrast to other lifelong learning competitors, which often exhibit a fragile balance between accuracy and fairness metrics, particularly under significant demographic distribution shifts.
> > > > > We greatly value your feedback and would be thankful for any additional comments or insights you might have.

---

> > > > > > ### Author Response · Authors · 2023-11-22
> > > > > > **Reference of the Latest Response to Reviewer E5F7**
> > > > > >
> > > > > > Reference:
> > > > > >
> > > > > > 1. Zhang, Brian Hu, Blake Lemoine, and Margaret Mitchell. "Mitigating unwanted biases with adversarial learning." In AIES, pp. 335-340. 2018.
> > > > > > 2. Chowdhury, Somnath Basu Roy, and Snigdha Chaturvedi. "Sustaining fairness via incremental learning." In AAAI, vol. 37, no. 6, pp. 6797-6805. 2023.
> > > > > > 3. Moyer, Daniel, Shuyang Gao, Rob Brekelmans, Aram Galstyan, and Greg Ver Steeg. "Invariant representations without adversarial training." In NeurIPS. 2018.
> > > > > > 4. Cerrato, Mattia, Marius Köppel, Roberto Esposito, and Stefan Kramer. "Invariant representations with stochastically quantized neural networks." In AAAI, vol. 37, no. 6, pp. 6962-6970. 2023.
> > > > > > 5. Louizos, Christos, Kevin Swersky, Yujia Li, Max Welling, and Richard Zemel. "The variational fair autoencoder." In ICLR. 2016.
> > > > > > 6. Liu, Ji, Zenan Li, Yuan Yao, Feng Xu, Xiaoxing Ma, Miao Xu, and Hanghang Tong. "Fair representation learning: An alternative to mutual information." In KDD, pp. 1088-1097. 2022.
> > > > > > 7. Hardt, Moritz, Eric Price, and Nati Srebro. "Equality of opportunity in supervised learning." In NeurIPS. 2016.
> > > > > > 8. Alghamdi, Wael, Hsiang Hsu, Haewon Jeong, Hao Wang, Peter Michalak, Shahab Asoodeh, and Flavio Calmon. "Beyond Adult and COMPAS: Fair multi-class prediction via information projection." In NeurIPS. 2022.

---

### Comment · Reviewer_an9h · 2023-11-19
**Review stop**

Hello everybody,

on Friday I had an emergency medical intervention on my eyes and I am now unable to read for the next few weeks. This message is also written with an assistance.

Overall, I cannot further contribute to the review process.

All the best

---

### Meta-Review · Area_Chair_Cubr · 2023-12-05

**Metareview:**

The paper proposes a method called RaRULi to sustain fairness of decisions made in an incremental learning setting. This approach ​​prioritizes simpler tasks and adjusts task schedules based on fair representations.

I recommend rejecting this paper for several key reasons:
- There are strong concerns among the reviewers that the paper is not well written and hard to follow.
- There are concerns with the evaluation with regards to other baselines: both since those do not rely on task labels and since misclassification due to the chosen notion of fairness may have strongly affected the results.
- The experiments are limited to tabular data.

**Justification For Why Not Higher Score:**

I recommend rejecting this paper for several key reasons:
- There are strong concerns among the reviewers that the paper is not well written and hard to follow.
- There are concerns with the evaluation with regards to other baselines: both since those do not rely on task labels and since misclassification due to the chosen notion of fairness may have strongly affected the results.
- The experiments are limited to tabular data.

**Justification For Why Not Lower Score:**

N/A

---

### Decision · Program_Chairs · 2024-01-16

Reject